# Optimization of ship hull forms by changing CM and CB coefficients to obtain optimal seakeeping performance

**Mohsen Khosravi Babadi[1], Hassan Ghassemi** [2]*

**1** Department of Maritime Engineering, Amirkabir University of Technology, Tehran, Iran, **2** Int. School of Ocean Science and Engineering, Harbin Institute of Technology, Weihai, China

* gasemi@aut.ac.ir

**Data Availability Statement:** All relevant data are within the paper.

**Funding:** The author(s) received no specific funding for this work.

## Abstract

Ship design involves optimizing the hull in order to enhance safety, economic efficiency, and technical efficiency. Despite the long-term research on this problem and a number of significant conclusions, some of its content still needs to be improved. In this study, block and midship coefficients are incorporated to optimize the ship's hull. The considered ship was a patrol vessel. The seakeeping analysis was performed employing strip theory. The hull form was generated using a fuzzy model. Though the body lines generated by the midship coefficient ($C_M$) and block coefficient ($C_B$) varied indecently, the other geometric parameters remained the same. Multi-objective optimization was used to optimize $C_B$ and $C_M$. According to the results of this study, these coefficients have a significant impact on the pitch motion of the patrol vessel as well as the motion sickness index. Heave and roll motions, as well as the added resistance, were not significantly influenced by the coefficients of $C_M$ and $C_B$. However, increasing the hull form parameters increases the maximum Response Amplitude Operator (RAO) of heave and roll motions. The frequency of occurrence of the maximum roll RAO was in direct relation with $C_B$ and $C_M$. These coefficients, however, had no meaningful impact on the occurrence frequency of other motion indices. In the end, the $C_B$ and $C_M$ coefficients were selected based on the vessel's seakeeping performance. These findings might be used by shipbuilders to construct the vessel with more efficient seakeeping performance.

## 1. Introduction

Seakeeping refers to a vessel's ability to navigate safely during long storms and withstand rough conditions at sea. Seakeeping performance can be described as the dynamic behavior of a vessel in wind, waves, and currents. Among the performance indicators, there are comfort, crew workability, and damage to the vessel and cargo resulting from slamming and green water, as well as the vessel's earning potential.

There are six degrees of freedom on each vessel, including three linear motions (surge, sway, heave) and three angular motions (roll, pitch and yaw). In the first steps of designing

**Competing interests:** The authors have declared that no competing interests exist.

**Abbreviations:** f, Frequency; Fr, Froude number; g, Gravity acceleration; k, Wave number; $\overline{T}$, Average wave period; $T_Z$, Zero crossing period; $T_P$, Peak wave period; $\theta_n$, Angular motion; $L_{CF}$, Longitudinal center of buoyancy; $L_{CB}$, Longitudinal center of buoyancy; $K_G$, Vertical center of gravity; $K_B$, Vertical center of buoyancy; $KM_T$, Transverse metacenter height; $KM_L$, Longitudinal metacenter height; $BM_T$, Transverse metacentric radius; $BM_L$, Longitudinal metacentric radius; $GM_T$, Transverse metacentric height; $GM_L$, Longitudinal metacentric height; $Z_n$, Linear motion; P, Possibility; s, Spectrum; Fr13, Vessel cross frame number.

new vessels, the determination of these operational motions, either analytically or non-analytically, can lead to the development of design conditions. Seakeeping performance is dependent upon the shape of the hull, which is designed by naval architects to achieve the highest level of performance. The sea-going ship that operates in open waters rarely sails in calm weather. On the contrary, ship's behavior at sea is often affected by waves and wind. When a ship navigates in a seaway, the ship's forward speed decreases, compared to that in calm sea, because of added resistance due to winds, waves, rudder angle, and so forth. The magnitude of added resistance is about 15–30% of calm-water resistance [1]. The immediate effect of waves is ship's motions and accompanying phenomena, such as accelerations. Ship accelerations, in turn, particularly vertical ones, impact on the human body and may cause motion sickness. The term "motion sickness", on ships known as sea sickness, is understood as a sickness due to ship motions that results in physical discomfort, with such symptoms as irregular breathing, nausea, vertigo, paleness and vomiting. In extreme cases a passenger or crew member has to be transferred to hospital. The actual reason for sea sickness is lack of conformity between different stimuli, eye signals and the labyrinth (inner ear), received by the human brain. People mainly suffer from sea sickness under deck, where the eye does not register any stimuli that the labyrinth would interpret as motion [2].

Due to the abovementioned issued, the optimization of the hull is one of the most important aspects of ship design in order to improve both safety and economic and technical efficiency for ships. Despite the fact that this problem has been studied for a long time and has achieved many significant results, some contents need to be improved in practical applications. Among these are the presentation and transformation of existing hulls, the development of an optimal mathematical model for a particular type of ship, or the development of a method for solving the objective function, etc.

Optimization of a vessel hull form includes a number of nontrivial issues, including selection of an appropriate function, choosing optimization scheme, geometrical representation of hull surface and choice of related design variables and constraints, selecting a practical and robust numerical tool for evaluating the objective function, and decision to perform optimization for a single point design or for multiple point design, e.g. for a single ship speed or for a range of speeds [3]. Different numerical schemes have been used for studying the ship hull form optimization problem. Some of the most relevant studies are listed in Table 1.

Maxsurf is a naval architecture software that provides integrated tools for hull modeling, stability, motions and resistance prediction as well as structural modeling. This software is a well-known numerical tool for ship analysis [53–57] that also used in recent ship optimization studies [58–60]. This software was also used in previous studies of the authors. Khosravi Babadi and Ghaseemi (2013) performed a number of numerical and experimental studies, including investigating the effect of variations in some geometrical parameters and hull form coefficients, including water plane coefficient ($C_{wp}$) and prismatic coefficient ($C_p$), on sea-

**Table 1. Relevant studies on ship hull form optimization.**

| Method | Conducted by authors |
|---|---|
| Thin-ship strip theory | [4–6] |
| Slender-ship approximation | [7] |
| Fourier-Kochin flow method | [8, 9] |
| Strip theory | [10–13] |
| Potential-flow panel methods based on Rankine sources | [14–23]. |
| Boundary element method | [24–30] |
| CFD | [31–52] |

keeping response [61], using multi-objective genetic algorithms optimization to optimize the vessel's body [62].

The Preference Ranking Organization METHod for Enrichment of Evaluations (PRO-METHEE) that is used in this study has particular application in decision making, and is used in a wide variety of decision scenarios, including portfolio and stock selection problems [63–65], environmental issues [66–68], energy management [69, 70] and shipping industry [71–75].

While hull form optimization has been studied extensively, the effects of changing main hull form coefficients such as $C_B$ and $C_M$ on motions have not been investigated. This paper fills that gap.

Although some recent studies have addressed the issue of ship hull optimization, more research is required to examine the effect of different hull form coefficients, including ship midship and block coefficients. The purpose of this study is to provide insight on how to optimize the ship hull taking into account these two coefficients. This study involves the following steps being performed:

1. Assessing the effect of the $C_B$ and $C_M$ parameters of the vessel on seakeeping indices (heave, pitch, and roll motions, added resistance, and motion sickness).

2. Optimizing the vessel's form based on seakeeping parameters.

The key novelties of this work are: (1) using block and midship coefficients as optimization parameters to improve seakeeping and (2) developing a fuzzy model to generate hull form variations while keeping other parameters constant.

Unlike previous hull form optimization studies that use complex geometrical parameters, we use only $C_B$ and $C_M$ as variables in a novel fuzzy hull generation model to provide useful insights.

In order to achieve the above-mentioned goals, a fuzzy model was developed in order to generate the hull form. The body lines generated by $C_M$ and $C_B$ in this model vary indecently, but do not affect the other geometric parameters ($C_P$,$L$ (Length) and $B$ (Breadth)). An index of seakeeping performance (SPI) is defined as an objective function expressing dynamic behavior. Using multi-objective optimization, some values of $C_B$ and $C_M$ were derived that optimize seakeeping.

It will be shown that by optimizing these coefficients, pitch and MSI will improve. On the other hand, the effect of these coefficients on the roll and heave motion as well as the added resistance is negligible.

## 2. The mathematical procedure

In this section, the mathematical procedure made to optimize the vessel hull form is presented.

### 2.1. Strip theory

The estimation of ship motions in the presence of regular waves, arbitrary heading, and constant forward speed of the ship, as well as the calculation of wave-induced horizontal and vertical shear forces, bending moments, and torsional moments, are based on "strip theory". According to the theory, the coefficients of the related ship in the ship motion equations in two dimensions are calculated, and then integrated throughout the ship length and transformed into the 3D global coefficients.

The basic idea of the strip theory is dividing the hull into several slices along the longitudinal direction as shown in Fig 1. Under given loading conditions and speed conditions, for any combination of wave frequency and wave direction, the hydrodynamic coefficients, such as additional mass, additional damping, Froude–Krylov wave force, and diffraction wave force,

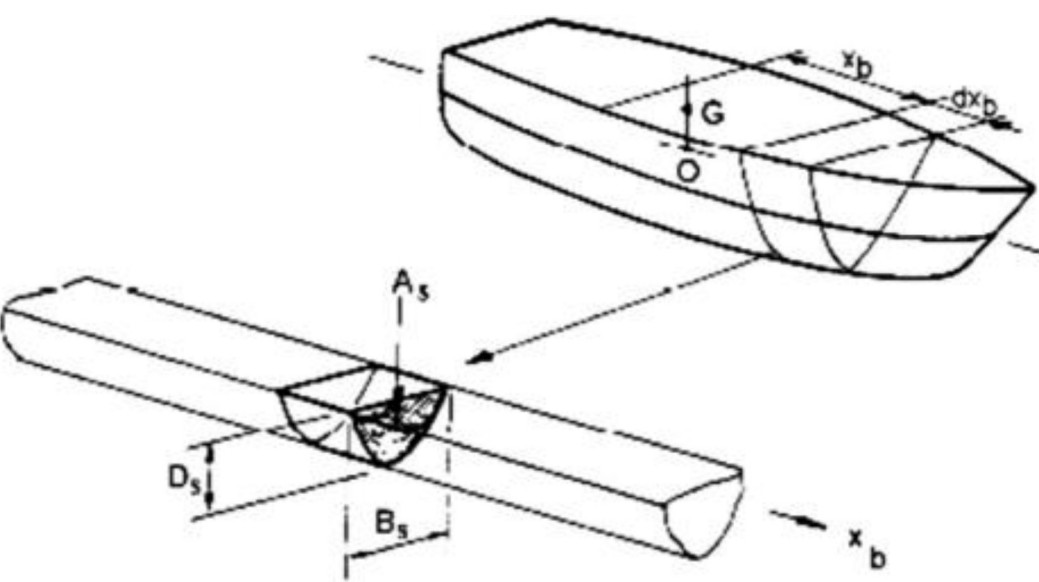

**Fig 1. Representation of strip theory by cross sections [76].**

were calculated on each slice by applying a unit amplitude regular wave to the hull. Finally, the force of each slice is integrated longitudinally to obtain the force of the entire hull. In a regular wave, a ship's movement may be broken down into two separate issues to solve.

1. Radiation issues: Only the ship's free swing motion is taken into account because there is no impact wave. This condition's hydrodynamic force is made up of words for increased mass force, damping force, and restoring force.

2. Diffraction problem: Only the impacts of regularly occurring incident waves on the hull are considered, assuming that the ship is stationary. Wave forces make up the hydrodynamic force at this moment. Incident wave force and diffraction wave force make up wave force. The latter is the wave force produced by the wave when it contacts the hull, whereas the former just takes into account the impact of the incident wave on the hull and ignores the impact of the hull's presence on the flow field. When the ship motion responses are linear and harmonic, the coupled six degree of freedom (6-DoF) equation of motion in the frequency domain is as follows:

$$\sum_{n=1}^{6}[(M_{jn} + A_{jn})\ddot{\eta}_n + B_{jn}\dot{\eta}_n + C_{jn}\eta_n] = F_j e^{i\omega_e t} \quad for\ j = 1, \ldots, 6 \tag{1}$$

where $M_{jn}$ and $A_{jn}$ are the generalized mass and added mass matrices, respectively, $B_{jn}$ and $C_{jn}$ are the damping and restoring coefficients. The hydrodynamic coefficients representing the jth degree of freedom caused b mentioning the nth degree of freedom. $F_j$ is the exciting force and moment. $\eta_n$, $\dot{\eta}_n$ and $\ddot{\eta}_n$ are the displacement, velocity and acceleration of the nth DOF, respectively. $F_j$ and $w_e$ are the forces (or moments) acted to the ship and wave encounter frequency.

The frequency domain transfer function of the hull motion is then calculated by substituting the hydrodynamic and wave forces into the equation for the six-degrees-of-freedom motion of the hull. The relative motion connection makes it clear that the motion response

may be acquired at any point along the hull and that time differentiation can be used to get the appropriate speed and acceleration. Then, the hydrodynamic and wave forces are substituted into the hull 6-DoF motion equation to obtain the frequency domain transfer function of the hull motion. It is known from the relative motion relationship that the motion response at any position of the hull can be obtained, and the corresponding speed and acceleration can be obtained by time differentiation. For the motion calculation, the total hydrodynamic coefficients were computed with the Salvesen–Tuck–Faltinsen (STF) strip theory that is well known, so the details are not described here.

**Response amplitude operator (RAO).** RAO is also known as the transfer function (similar to the response curve of an electronic filter), describes how the response of the vessel changes with frequency. These are usually dimensionless due to the height of the waves and the slope of the waves. RAOs tend toward unity at low frequencies; this is where the ship simply moves up and down in waves and acts as a stopper. At high frequencies, the response tends to zero due to the effects of many destructive micro waves along the length of the ship. Usually, ships will also have larger peaks than unity; this occurs near the natural period of the circuit. The peak is due to resonance. An RAO value greater than unity indicates that the ship's response is greater than the amplitude (or slope) of the wave.

**Motion response theory in irregular waves.** We typically believe that the response of a ship's linear system may be superimposed homogeneously when predicting a ship's seakeeping performance in irregular waves. Additionally, the output is treated as a stationary random process when the input is a stationary random process. Under such assumptions, hydrodynamic calculation also known as the transfer function, may be used to determine the relevant connection between the response variable and the wave frequency (or period, wavelength) for each wave direction, each wave speed, and each loading condition (or response amplitude operators). The following formula may be used to get the response spectral density function from the transfer function and wave spectral density function:

$$S_R(\omega, \beta, H_S, T_Z, U, C) = H^2(\omega, \beta, U, C).S(\omega, H_S, T_Z), \tag{2}$$

where $H(w,\beta,U,C)$ is the response amplitude operator transfer function, $S_R$ is the response spectral density function, $\beta,U,C$ represents the heading angle, ship speed, and ship loading condition, respectively. $w$; $H_S$; $T_Z$ are the wave frequency (rad/s), significant wave height (m), and average zero-crossing period (s), respectively. The encounter frequency will alter with the heading angles as the ship moves across the waves. The link between the encounter frequency and the wave frequency is as follows because the wave spectral density function at the encounter frequency and the wave spectral density function at the wave frequency have the same amount of energy [76]:

$$\omega_e = \omega - \frac{\omega^2}{g}.U.\cos(\beta) \tag{3}$$

The motion variance is given by the area under the motion energy spectrum as

$$m_{o_e} = \int_0^\infty S_r(\omega_e).d\omega_e. \tag{4}$$

Hence, the $\sqrt{m_{o_e}}$ represents the RMS motion, and the significant motion amplitude is twice the RMS motion. In addition, the RMS velocity and acceleration are given by $\sqrt{m_{2_e}}$ and $\sqrt{m_{4_e}}$ [76].

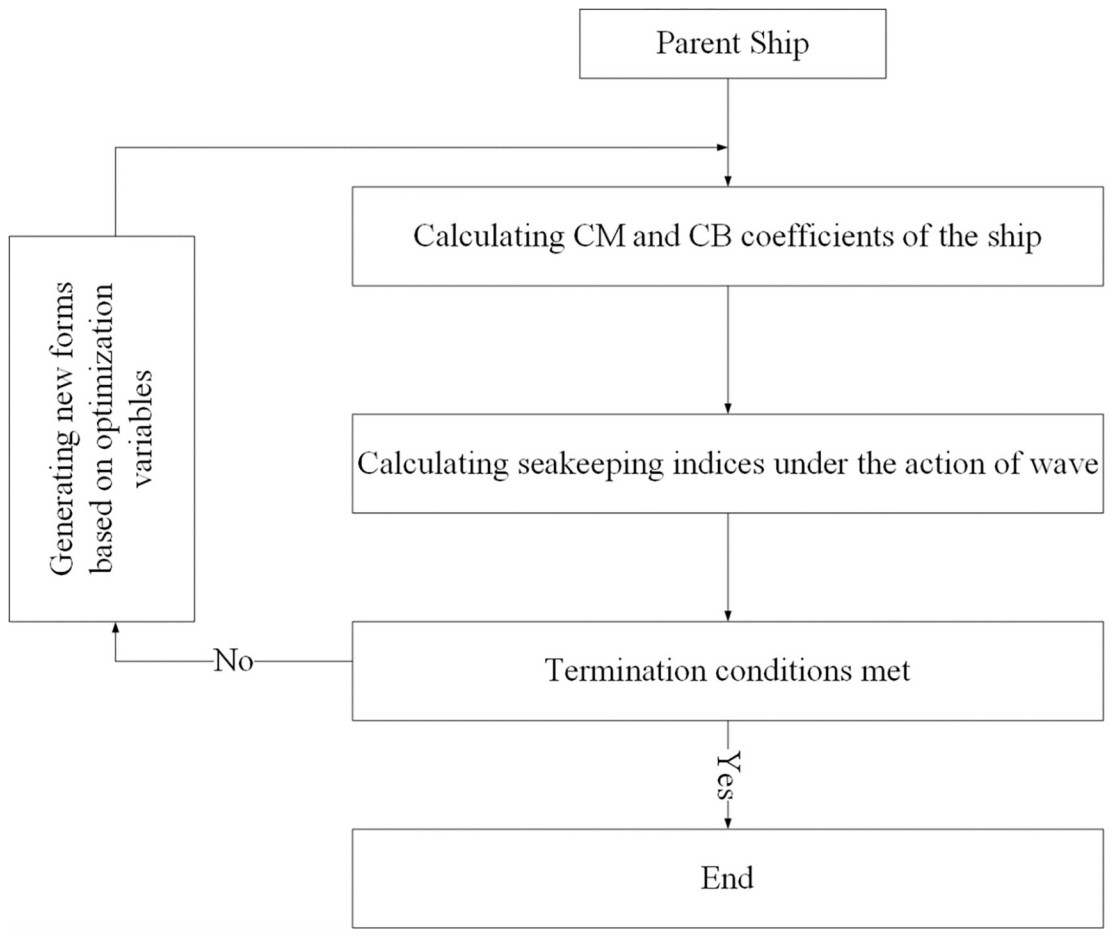

**Fig 2. Flowchart of the optimization procedure.**

**Motion sickness index (MSI).** Despite scientific observations and research, no exact relations have been determined between ship motions and motion sickness. McCauley and O'Hanlon estimated quantitatively the impact of ship motions on the percentage of people that would suffer from sea sickness. It turned out that vertical accelerations in particular were responsible for motion sickness, while rolling and pitching had slight influence. Additionally, it was found that at a frequency of 0.167 Hz the occurrence of motion sickness increased. The Motion Sickness Incidence (MSI) index is commonly used for assessing possible occurrence of the illness:

$$MSI = 100 \left[ 0.5 \pm erf \left( \frac{\pm log_{10} \frac{a_v}{g} \pm \mu_{MSI}}{0.4} \right) \right] \tag{5}$$

where MSI is the motion sickness incidence index, erf is the error function, $a_v$ is the mean value of vertical accelerations at a selected point and $\mu_{MSI} = -0.819 + 2.32(log_{10} w_E)^2$.

**Optimization procedure.** The flowchart of the optimization procedure is given in Fig 2.

## 2.2. Modeling fuzzy changes in desired parameters

Fuzzy logic has advantages in ship hull optimization, including:

- Handling imprecise and uncertain data: Fuzzy logic effectively deals with imprecise and uncertain information, common in ship design and optimization.

- Flexibility: Fuzzy logic incorporates subjective human knowledge and expertise into the optimization process, making it adaptable to different design requirements.

- Non-linear relationships: Fuzzy logic models complex, non-linear relationships between design parameters and performance criteria, which traditional optimization methods may struggle with.

- Robustness: Fuzzy logic-based optimization methods are often more stable, handling variations and uncertainties in design parameters without significantly impacting results.

This paper presents a fuzzy structure model for midship coefficient changes. The variation of these coefficients does not affect the other geometric parameters. Changes should be applied in such a way as not to affect the volume of the vessel. It is, therefore, necessary to develop a mathematical model of the changes that will cause the exact change in the desired parameter while maintaining other geometric parameters constant while holding constant the volume.

A fuzzy membership function, which is an extended Bell membership function, is defined as a function that makes changes to the body's lines. As a result, if any other changes are made, it increases and decreases the same volume, thus maintaining a constant volume change. Bell function is expressed as follows and its shape is a modified Gaussian distribution. Fig 3 illustrates the distribution shape of the Bell function.

$$Bell(x; a, b, c) = \frac{1}{1 + \left| \frac{x-c}{a} \right|^{2b}} \tag{6}$$

Fig 3 illustrates how the fuzzy function at the upper level makes good changes to the body line in the middle and returns the variant body lines to the original lines at both ends with instant cuts in both directions, with the bell acting as a coefficient of variation. It is now necessary to restore the taken area of one side to the opposite direction using the same function used in the interval (0, -1). Therefore, this function is multiplied by a sine function. The fuzzy

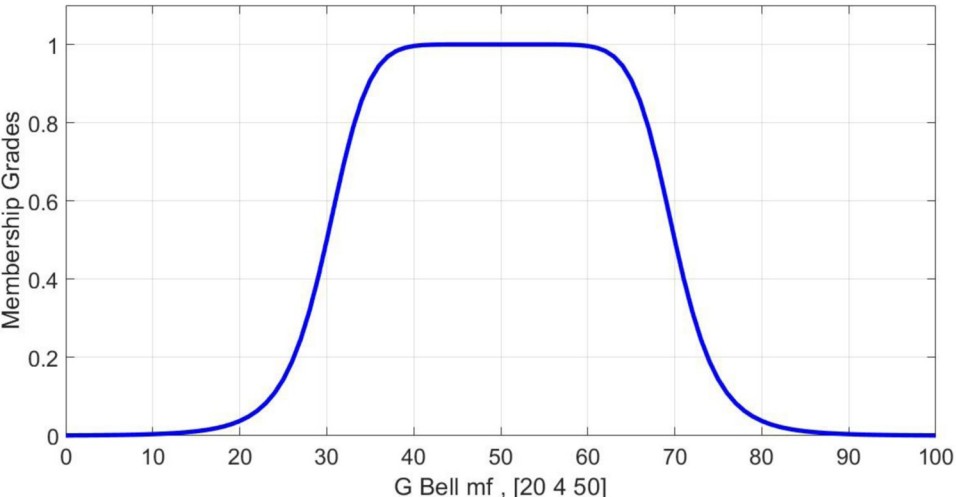

**Fig 3. Bell's membership functions.**

function coefficients in this model are defined as follows:

$$G(x) = gbellmf(x1, [340])$$

$$a = 3, b = 4, c = 0. \tag{7}$$

The following is a description of the effects of parameters on the function:

Coefficient $b$: Coefficient $b$ is a positive number and is usually considered to be 4. The number corresponds to the upper part of the curve and contributes to the flat portion of it. This coefficient has a significant impact on the performance of the function. Fig 1 illustrates the changes in the crude bell membership function.

Coefficient $a$: Based on mathematical calculations and modeling, a coefficient equals 1.44 times the length of each line within a body. There will obviously be a difference in the coefficient of the body lines of the vessel.

Coefficient $c$: It determines the center of the bell function that is considered equal to the half-length of each line.

The sine function of the fuzzy function multiplied by Bell is given below:

$$F(x) = \sin\left(\frac{x - \min(x)}{\max(x) - \min(x)}\right) \tag{8}$$

Data set x represents the length of the body lines to a point at which the fuzzy shift applies the same area to the other side. The length of the new points is determined by the function $H(x)$. Using coefficient $L$ at the end of the function, the different sizes of the changes are applied manually. This is the fuzzy function - neurological function obtained from Bell's model to apply to the vessel model:

$$H(x) = [G(x) \times F(X) \times L]\left[\frac{x}{\max\max(x) - \min(x)} + \min(x)\right]$$

$$H(x) = \left[\left[\frac{1}{1 + |\frac{x}{3}|^8}\right] \times \left[\sin\left(\frac{x - \min(x)}{\max\max(x) - \min(x)}\right)\right] \times L\right] \\ \times \left[\frac{x}{\max\max(x) - \min(x)} + \min(x)\right] \tag{9}$$

Using the obtained membership function, a fuzzy inference system is formed using a membership function $H(x)$ and a simple inference function. Fuzzy relations are applied to any of the vessel body lines based on the customization done. As a result of using the above model, the vessel volume is not altered and the coefficient $C_B$ and $C_M$ are affected positively by adjusting geometrical values.

## 2.3. The objective functions in the frequency domain

Based on the original model, 10 variant models are generated for coefficients $C_B$ and $C_M$ and seakeeping parameters which include roll, pitch, heave, added resistance, and motion sickness acceleration calculated at three heading angles. For an irregular wave, the dynamic behavior of a vessel is calculated using the strip theory method. As the coefficient changes are very small, the results of calculations are very similar for different forms of the body. Some wave frequencies, however, demonstrate differences in dynamic behavior. Most of the difference can be found in the areas of maximum and minimum amplitudes. As a result, the objective of optimization is to reduce maximum amounts, thereby reducing the effective amounts of seakeeping

parameters. Five objective functions, including roll, pitch, heave, seasickness acceleration, and added resistance in the waves, are plotted. As a result of the independent solution of the roll equation in the calculation, we will not have roll for head waves. For $C_B$ and $C_M$, each diagram shows a variation range of ±3%.

In multi-objective optimization with seakeeping purposes, for each generated geometric model and every heading angle, five RAO curves will be generated based on the seasickness accelerations. These curves will include roll, pitch, heave, added resistance, and MSI. The optimization does not take into account a certain frequency; the significant value for every curve or parameter within a certain frequency range will be calculated. For example, $C_B$ and $C_M$ coefficients with percentage changes of +3%, +2.5%, 2%, 1.5%0%, -1.5%, -2%, -2.5% and -3% and heading angle of 150 degrees can be considered nine significant for each parameter.

For the purpose of achieving the objective functions, polynomial curve fitting is used, contributing these nine points. The objective function can be applied to each coefficient (variable) at each heading angle. Each heading angle should be combined with weights for any parameter (such as roll) to determine the objective function.

## 2.4. Wave spectrum and motion indices spectrum

The vessel is planned to perform operation in the Gulf of Oman. For this sea, the Pierson Moskowitz wave spectrum is suitable to be used as the wave spectrum. Generalized Pierson Moskowitz spectrum is as follows:

$$S(f) = \frac{A}{f^5} \exp(-B/f^4) \tag{10}$$

In this study, we used ITTC spectrum. For ITTC spectrum the input parameters are $H_S$ and one of the $T_E$, $T_p$, $\bar{T}$ or $T_Z$. Also, $A$ and $B$ parameters are defined as follows:

$$A = \frac{0.0081}{K^4} g^2$$

$$B = \frac{0.0081}{K^4} \frac{4g^2}{H_s^2}$$

$$K = \frac{T_E}{2.137} \sqrt{\frac{g}{H_s}}, K = \frac{T_p}{2.492} \sqrt{\frac{g}{H_s}}, K = \frac{\bar{T}}{1.924} \sqrt{\frac{g}{H_s}}, \text{ or } K = \frac{T_z}{1.771} \sqrt{\frac{g}{H_s}} \tag{11}$$

Pierson-Moskowitz table for sea states (Beaufort) 3 and 5 are listed in Table 2:

RAO is the relation between the vessel motions to the wave amplitude which is usually draws in the shape of a dimensionless diagram based on the incidence frequency or dimensionless frequency. For linear and angular motions, RAO is defined by Eqs 16 and 17,

**Table 2. Pierson Moskowitz wave characteristics for different sea states.**

| Average wave period (s) | Significant wave height (m) | Sea state |
|---|---|---|
| 4.3 | 1.4 | Beaufort 3 |
| 6.4 | 3.2 | Beaufort 5 |

respectively:

$$RAO_n = \frac{Z_n}{\varsigma_a} = \sqrt{\frac{S_{Z_n}(\omega)}{S_{\varsigma_a}(\omega)}} \quad n = 1, 2, 3 \tag{12}$$

$$RAO_n = \frac{\theta_n}{K\varsigma_a} = \sqrt{\frac{S_{Z_n}(\omega)}{K^2 \times S_{\varsigma_a}(\omega)}} \quad n = 4, 5, 6 \tag{13}$$

where $n$ is the desired degree of freedom and $K$ is the wave number.

## 2.5. Mathematical modeling of the main body

This step involves entering the body line points into the MATLAB database. The training is carried out using a neural network with a transfer function in the hidden layer and a linear transfer function in the output layer, and 29 neurons in the hidden layer with a training function.

**Levenberg-Marquardt algorithm.** This Levenberg-Marquardt algorithm combines gradient reduction algorithms with Gauss-Newton algorithms (GNA). Unlike the Gauss-Newton algorithm, the Levenberg-Marquardt algorithm often finds a solution even if it begins far from the final minimum. Updating the parameters of this algorithm is performed by the following formula:

$$H(w) = J^T(w)J(w) + \mu I \tag{14}$$

and gradient:

$$\nabla F(w) = \sum_{n=1}^{N} \nabla E(w, n) = \sum_{n=1}^{N} \varepsilon^T(w, n).J(w, n) = e^T(w)J(w) \tag{15}$$

where:

$$e^T = scan(D - Y) = [\varepsilon^T(1) \ldots \varepsilon^T(N)$$

$$J(w) = \begin{bmatrix} J(1) \\ \vdots \\ J(N) \end{bmatrix} \tag{16}$$

In order to update the weights, the following equation is used:

$$\Delta w = -\nabla F.H^{-1} = -e^T(w)J(w)(J^T(w)J(w) + \mu I)^{-1} \tag{17}$$

Jacobian matrices are calculated using the same procedure as gradient matrices, except that derivatives are used instead of differences. The algorithm of this method, assuming matrix X is the input matrix, is as follows:

1. The following matrices are calculated:

- H secret signal

- Y output signal

- Derivatives of two-layer activator functions $\emptyset$ and $\varphi$

2. For each hidden layer and output layer, calculate the Jacobian matrix:

$$J^h = -S^h \bigotimes X^T, \text{ where } S^h = diag(\varphi').W^y.diag(\psi')$$

$$J^h = -diag(\varphi') \otimes h^T \tag{18}$$

3. By choosing the μ value, the weight change is calculated:

$$\Delta w = -\nabla F.H^{-1} \tag{19}$$

4. Upon determining the amount of performance change, the error is checked, and if it increases, μ is decreased until the error decreases.

5. The weights have been updated and we have returned to the first stage.

## 2.6. Seakeeping and seakeeping performance index (SPI)

The science of seakeeping involves investigating and predicting vessel motions. Designers are required to provide information regarding the vessel's seakeeping performance, including local motions and accelerations, added resistance, deck wetness, and bow slamming. "Vessels suitable for seagoing" must be capable of maintaining their motion in harsh conditions, so that deck wetness does not occur (Fig 4). The vessel should be independent of wind or wave direction, continue to follow its intended course even in inappropriate locations, and be able to quickly adjust to small angle changes. In addition, it must maintain a steady speed without slamming or abnormal fluctuations in the torque of the power transmission axis. Motions of the ship must be within an acceptable range and vibrations should not be excessive.

The Seakeeping Performance Index (SPI) is a common measure of how well a ship handles rough seas. It calculates the percentage of time that the ship stays within specific motion limits. The SPI depends on assumptions about the frequency of different sea states and the likelihood of different ship speeds and headings. To evaluate the SPI, we predict how the ship will move for each combination of heading, speed, and sea condition. We then compare these predictions to a set of criteria that determine the optimal performance limits for the ship's mission in that particular sea condition.

## 2.7. Simulation procedure

For changes to be applied to one parameter and other coefficients to remain constant, changes should be applied to the area under the draft, and anti-changes should be applied to the same area of the body lines. This will lead to the rise and fall of the center of gravity and subsequently the change of KB. For this purpose, the focus of the modeling is placed on the front lines of the vessel, which are called "tearing" lines. The modeling steps are introduced by the following steps:

1. Using the cross sections of the vessel's front lines, a curved line is selected.

2. Body line Fr13 divides the desired float into the upper part of the chine and the lower chine, each of which is regulated using a polynomial matching method. Based on the results, we have obtained the following relationships:

$f(x) = 2.167x - 3642$ above the chain

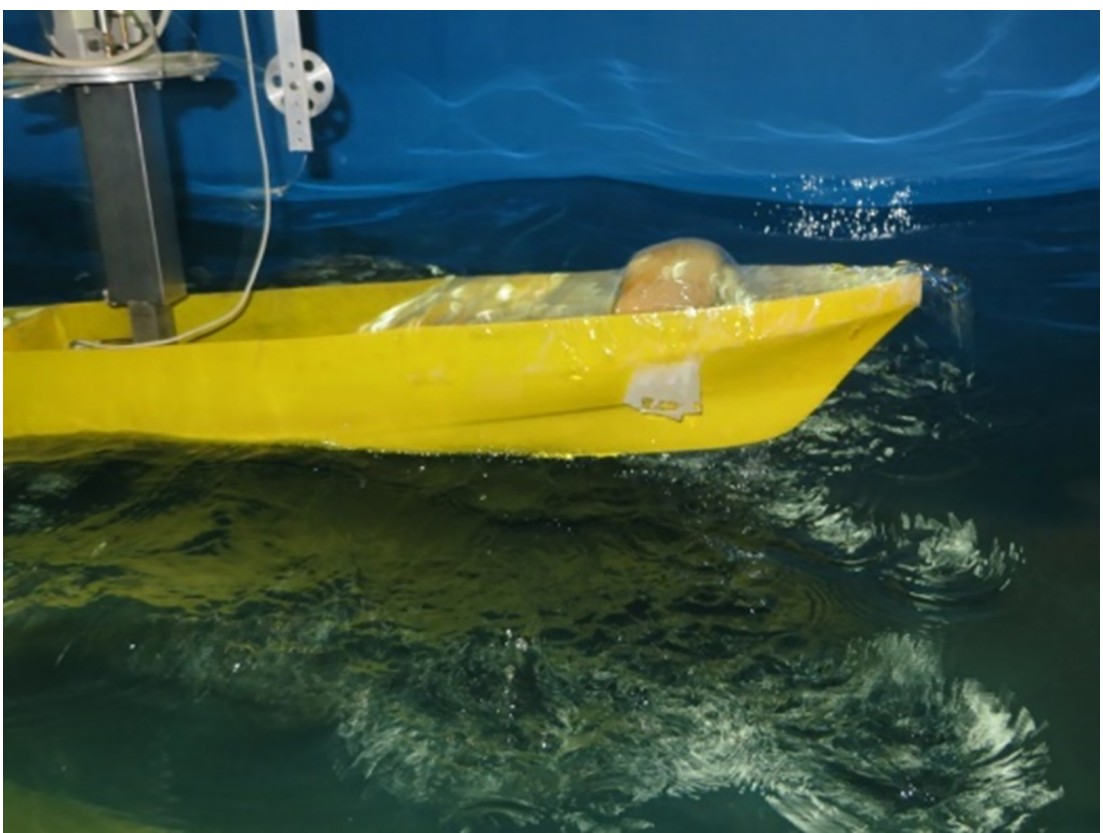

**Fig 4. Deck wetness in the model and harsh sea (waves with a height of 3.0 cm).**

$f(x) = 0.0002008x^2 + 0.3912x + 63.19$ bottom of chain

1. As a result of the trial-and-error method, we obtain a calculated model that is equal to the displacement area.

2. Polynomial matching is performed separately for each part of the modeled graph.

3. Steps 1 to 4 are repeated for the other four body lines.

   Fig 5 shows the modeled body lines.

## 2.8. Effective seakeeping parameters

This study investigates the dynamic effects of the vessel in the wave, as well as the operability criteria according to the type of operation and sub-systems required by the vessel. The roll, pitch, yaw, seasickness acceleration, and added resistance at the headings of 120, 150, and 180 are of the utmost importance. In addition to satisfying many criteria, by reducing the fluctuation range of these parameters, the derivatives (speed and acceleration) of these parameters are also reduced. These parameters will be used to determine the seafaring performance index.

   **RAO and effective seakeeping parameters.**   Following the process described in the previous sections, geometric coefficients were determined, and based on the range considered (±3%), 64 models were produced. In this section, we will calculate the relationship between the effective seakeeping parameters and the geometric coefficients. This means that when considering the effective seakeeping parameters, such as roll, pitch, heave, seasickness acceleration

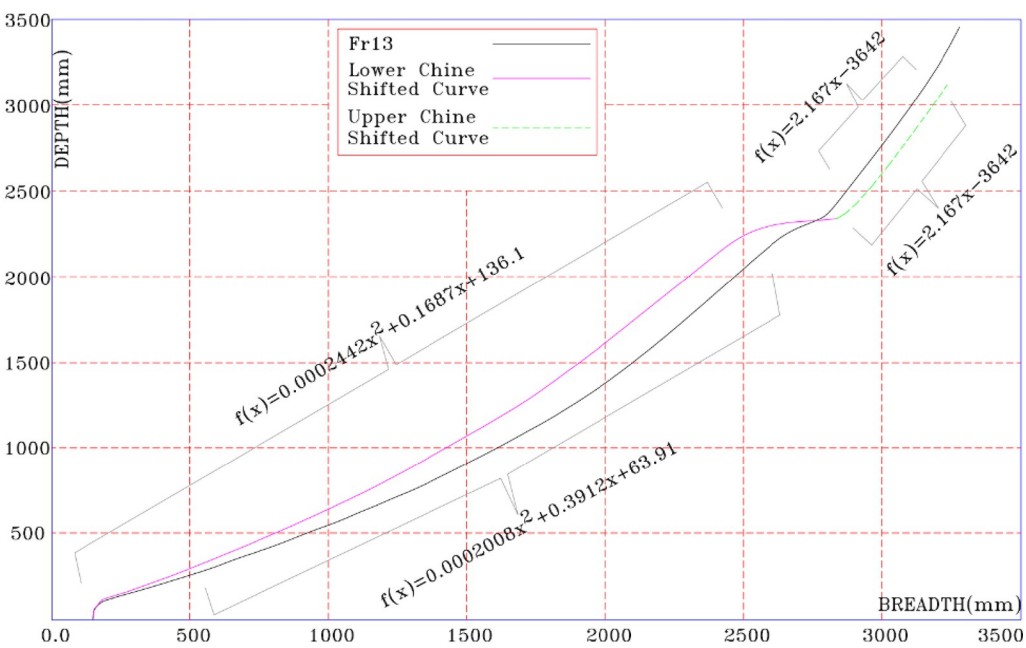

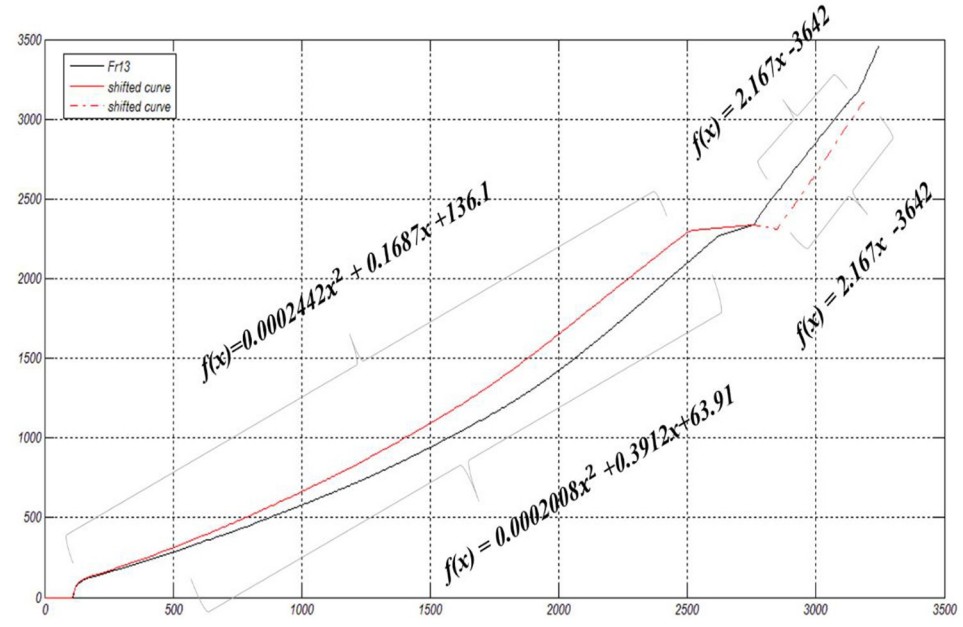

**Fig 5. Modeled body lines.**

and added resistance at different wave impact angles, the goal is to obtain these parameters according to geometric coefficients. The following steps will be followed:

1. Calculation of the RAO spectrum for each coefficient in all three incident angles.
   There is a RAO spectrum diagram for each parameter (e.g., Heave motion), each incidence angle, and each coefficient (e.g., $C_B$). Note that for the parameters of roll, pitch, heave and

added resistance, the amplitude of wave excitation is used to calculate RAO. There is no RAO associated with motion sickness acceleration.

2. RMS calculation for each RAO spectrum curve.

3. Calculating effective parameters in terms of coefficients at each angle of incidence by normalizing the values.

**An example of $C_M$ and $C_B$ modifications.** It is necessary to change the points of each of the lines in order to apply the desired changes to the vessel's lines. We apply the changes in the set of points in the desired direction and analyze the results as a new modeled vessel. In this case, the systematic changes are applied in accordance with the appropriate fuzzy function. In this problem, the constant weight of the vessel is assumed, and the change in the variable is the $C_M$ coefficient.

According to the variable relationship of the $C_M$ coefficient, the goal is to maintain the numerator while changing the denominator. Under the waterline of the main body, for example, an area of the lower half of the line should be tilted to the right, and an equivalent area of the upper half should be tilted to the left. These changes will result in an increase in the output of the fraction's denominator and a decrease in the coefficient.

An example of the output of the proposed method for $C_B$ modeling is shown in Fig 6. This figure shows the main lines of the body in blue and a designed model in red.

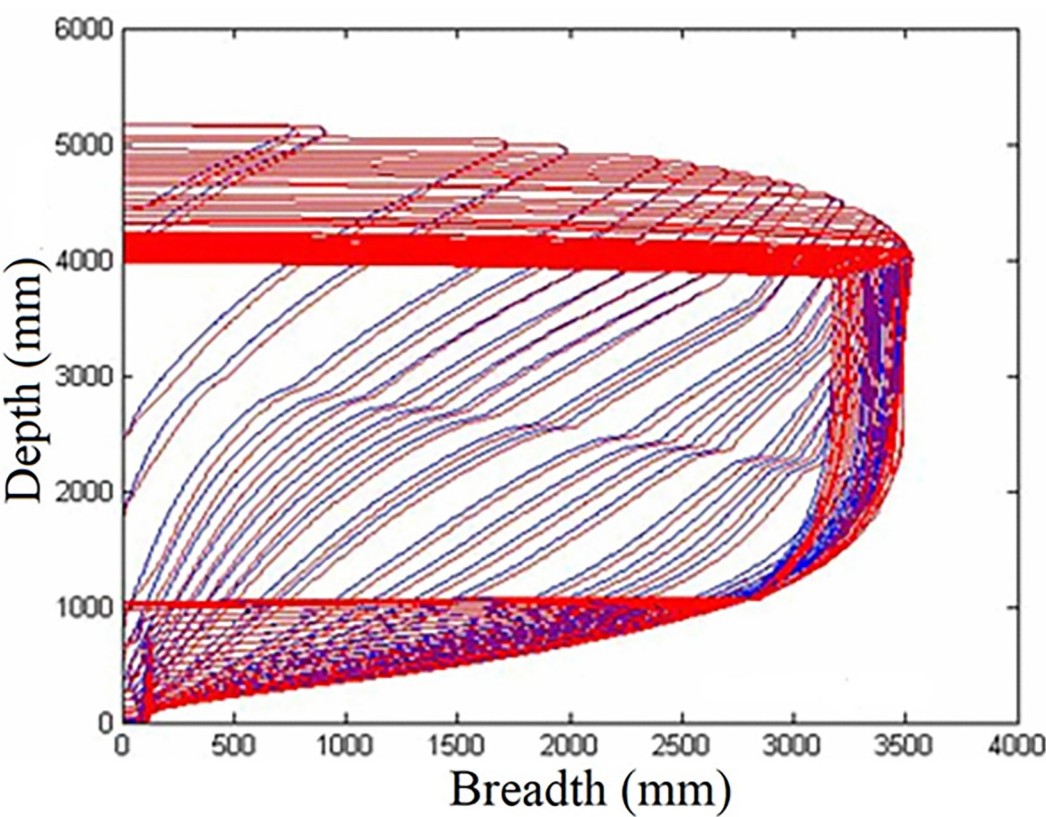

**Fig 6.** Designed model of $C_B$ simulation.

## 2.9. Choosing the best answer with fuzzy-promethea decision-making approach

In order to select the optimal values, the fuzzy-promethea approach is used. This method is a fuzzy multi-criteria decision-making method that performs the evaluation process by using the ranking of weighing coefficients. The ranking of options is done by comparing the pairs of options in each index. The comparison is measured based on a pre-defined superiority function with range [0,1]. The superiority function p, for comparing two options a and b in terms of index j, is defined as follows:

$$P_j(a, b) = P_j[d_j(a, b)] \tag{20}$$

where $d$ is the superiority function in the promethea method. Six most common superiority functions are shown in Fig 7.

The final ranking with the priority of the two options is obtained by summing the priority of all indicators, which is called the final ranking or overall value and is obtained by the following relationship:

$$\pi(a, b) = \sum_{j=1}^{k} w_j p_j(a, b), \quad \left(\sum_{j=1}^{k} w_j = 1\right) \tag{21}$$

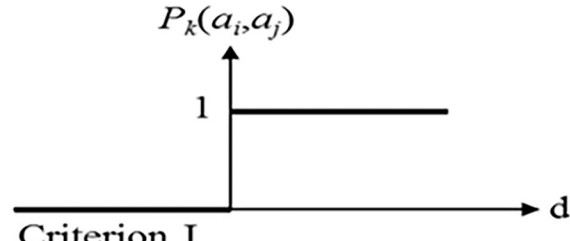

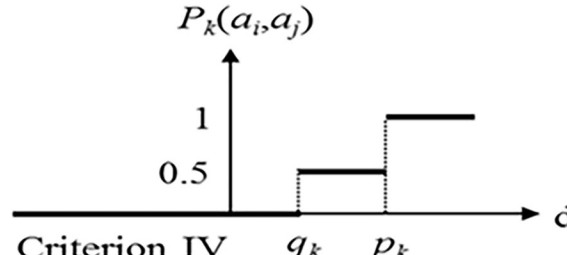

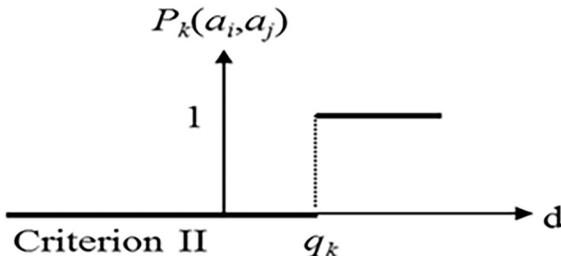

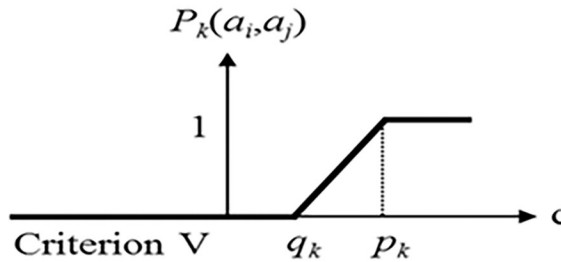

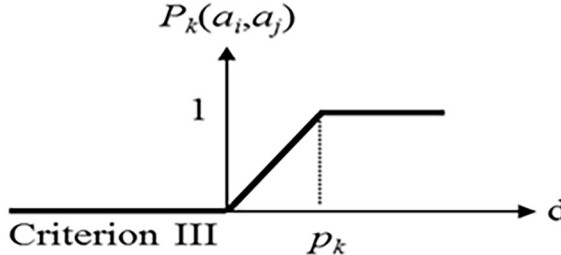

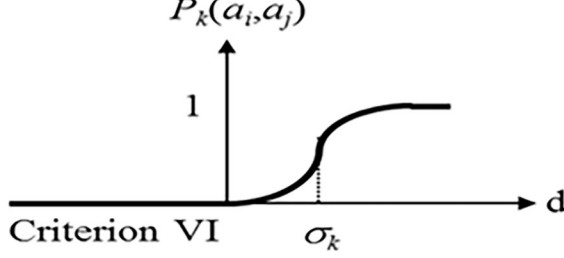

**Fig 7. The most common superiority functions in the promethea method.**

**Table 3. Main dimensions of the vessel.**

| Parameter | Dimension | Unit |
|---|---|---|
| Overall length ($L_{ov}$) | 38.5 | $m$ |
| Length between perpendicular ($L_{BP}$) | 36.5 | $m$ |
| Maximum width (B) | 7.5 | $m$ |
| Height (D) | 3.8 | $m$ |
| Displacement (Δ) | 300.3 | $ton$ |
| Draft (d) | 2.95 | $m$ |
| Vertical center of gravity (VCG) | 2.23 | $m$ |
| Longitudinal center of gravity from midship (LCG) | 2.33 | $m$ |
| Cruise speed | 15 | $Knots$ |
| Maximum speed | 35 | $Knots$ |
| Block coefficient ($C_B$) | 0.464 | - |
| Midship coefficient ($C_M$) | 0.672 | - |

where $w_j$ is the weight of the j-th index. If the number of options (n) is more than two, the final ranking is obtained by summing the values of pairwise comparisons. This also specifies the ranking order (input / output / final):

$$\emptyset^+(a) = \frac{1}{n-1} \sum_{x \in A} \pi(a, x) \tag{22}$$

$$\emptyset^-(a) = \frac{1}{n-1} \sum_{x \in A} \pi(x, a) \tag{23}$$

$$\emptyset(a) = \emptyset^+(a) - \emptyset^-(a) \tag{24}$$

## 3. An overview of the ship under consideration

A single-hulled patrol vessel with a V-shaped hull is being considered, which moves at a speed of 35 knots. The main specifications of the ship can be found in Tables 3 and 4. These tables may be amended as long as the specifications and body shape have not been finalized. A 3D view of the vessel is shown in Fig 8.

### 3.1. Validation

In this section, the results obtained using numerical model are compared with the laboratory data. A frame of the experimental study is shown in Fig 9. The wave amplitude is 1.5 cm. In this case, model presents more extreme motions in higher sea states. When the wave length is almost the same of the vessel length, the biggest heave and pitch motions are observed. This is evident in Figs 10 and 11. As these figures show, there is a suitable correspondence between

**Table 4. Hydrostatic characteristics of the ship hull.**

| $L_{CB}$ from AP ($m$) | 22.124 | $L_{CF}$ from AP ($m$) | 21.720 |
|---|---|---|---|
| $K_B$ ($m$) | 1.245 | $K_G$ fluid ($m$) | 2.3 |
| $BM_T$ ($m$) | 2.492 | $BM_L$ ($m$) | 97.958 |
| $GM_T$ ($m$) | 1.533 | $GM_L$ ($m$) | 97 |
| $KM_T$ ($m$) | 3.737 | $KM_L$ ($m$) | 99.204 |

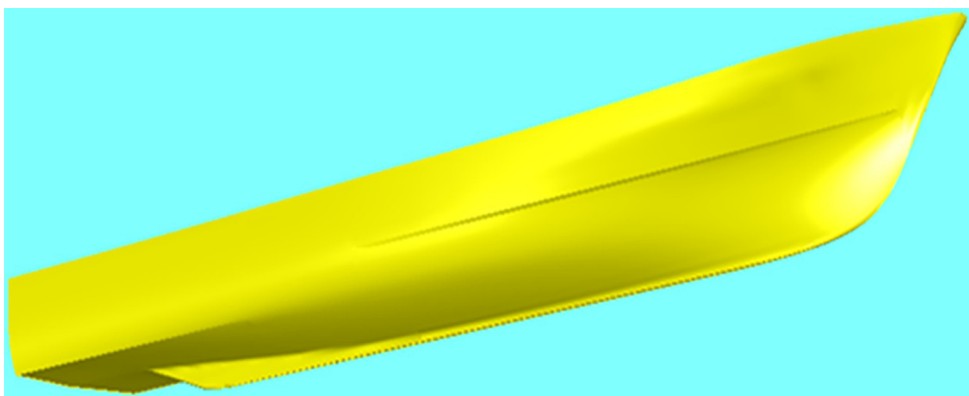

**Fig 8. 3D representation of the ship's hull form.**

experimental and numerical data. Hence, the numerical model is accurate enough to be used in further numerical investigations.

## 3.2. The initial state of seakeeping of the vessel under consideration

This section presents the response spectrum at three headings of 120, 150, and 180 degrees at 15 knots for a patrol vessel.

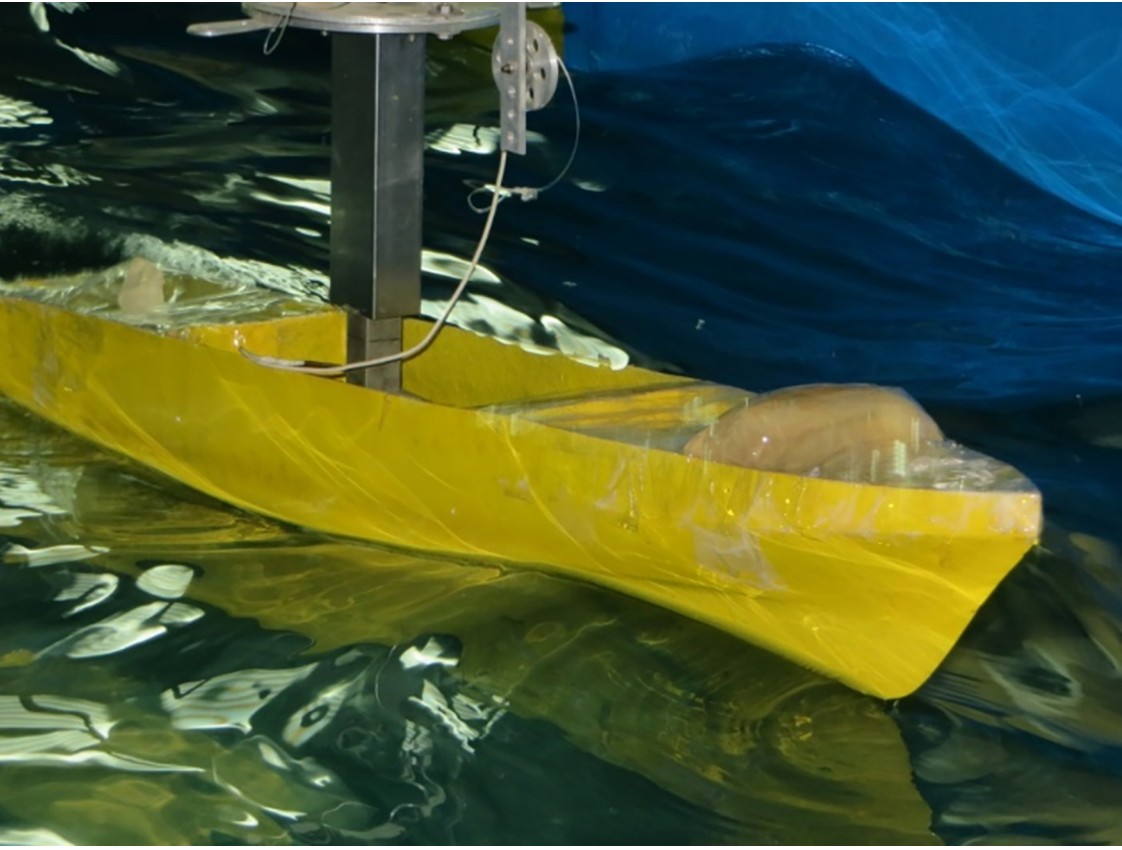

**Fig 9. The vessel moving at Fr=0.21 in head waves.**

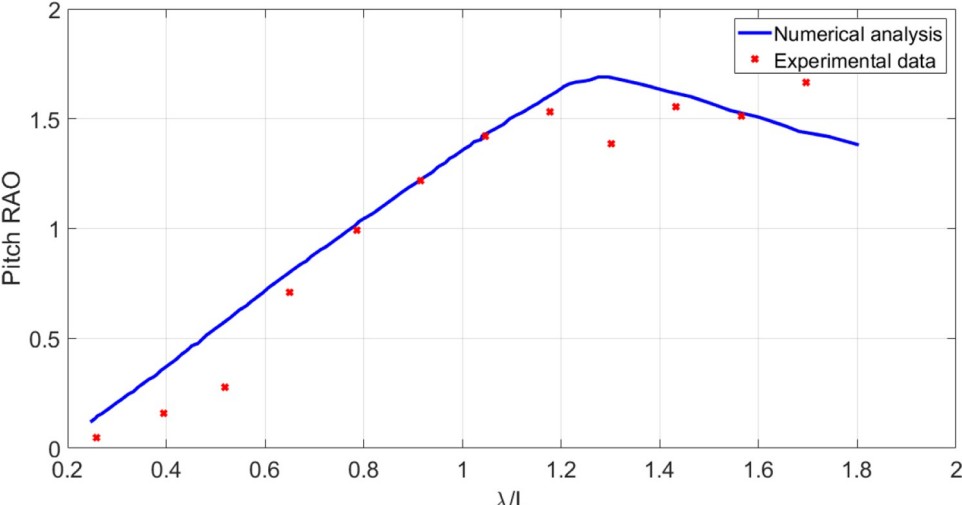

**Fig 10. Pitch RAO of the original model at Fr=0.21 in head waves.**

**Incidence spectrum.** For the considered vessel, the wave incidence spectrum at a speed of 15.0 knots and three different heading angles of 120, 150, and 180 degrees is shown in the Fig 12.

**Roll motion of the original vessel.** A summary of the vessel's roll motion is presented in the Fig 13. As can be seen, no roll motion is observed for headings of 180 degrees.

**Pitch motion of the original vessel.** Fig 14 presents the amounts of pitch motion spectrum of the vessel.

*Heave motion of the original vessel.* Fig 15 presents the amounts of heave motion spectrum experienced by the vessel at three different heading angles.

*Added resistance of the original vessel.* The added resistance of the vessel at different incidence angles is shown in Fig 16.

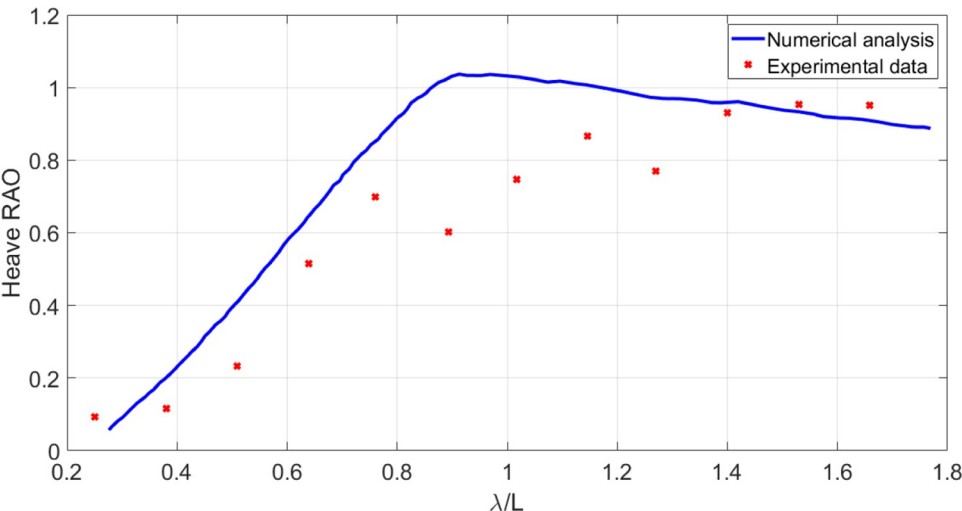

**Fig 11. Heave RAO of the original model at Fr = 0.21 in head waves.**

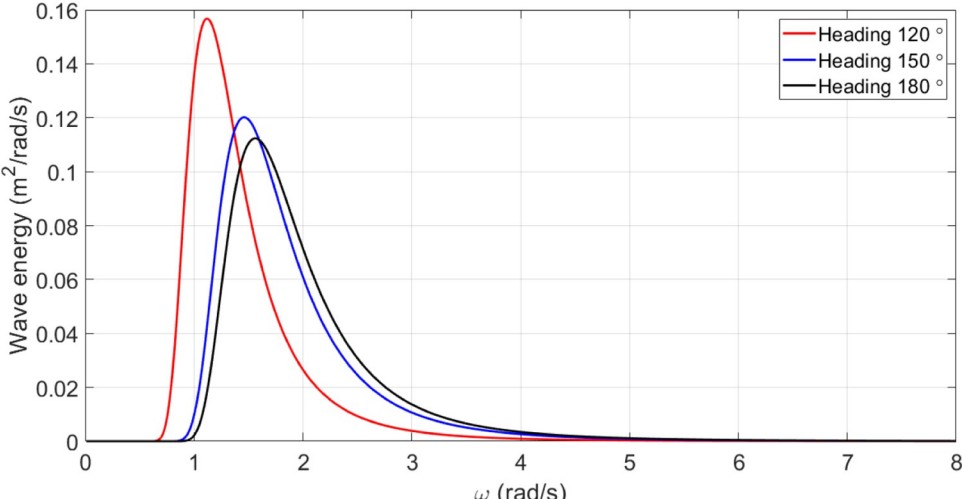

**Fig 12.** Incident wave spectrum for U = 15 knots for the original model of the vessel.

## 4. Results and discussion

### 4.1. Spectrum of effective seakeeping parameters for $C_M$ coefficient

In the single parameter $C_M$ spectrum, RAO diagrams were calculated for heave, pitch, roll motions, added resistance, and MSI against the incidence frequency at a speed of 35 knots for different headings. Figs 17–21 illustrate the RAO diagrams for heave, pitch, and roll motions, as well as added resistance and MSI. Figs 22–24 illustrate the RMS diagrams for heave, pitch, and roll motions.

The Fig 17 illustrates the heave RAO for changing CM coefficients at different wave frequencies. There was no significant impact of $C_M$ on heave RAO. However, the maximum heave RAO was increased by increasing the $C_M$. In contrast, decreasing $C_M$ resulted in a decrease in the maximum heave RAO. Further, the frequency of occurrence of the maximum heave RAO was not significantly affected by changes in $C_M$.

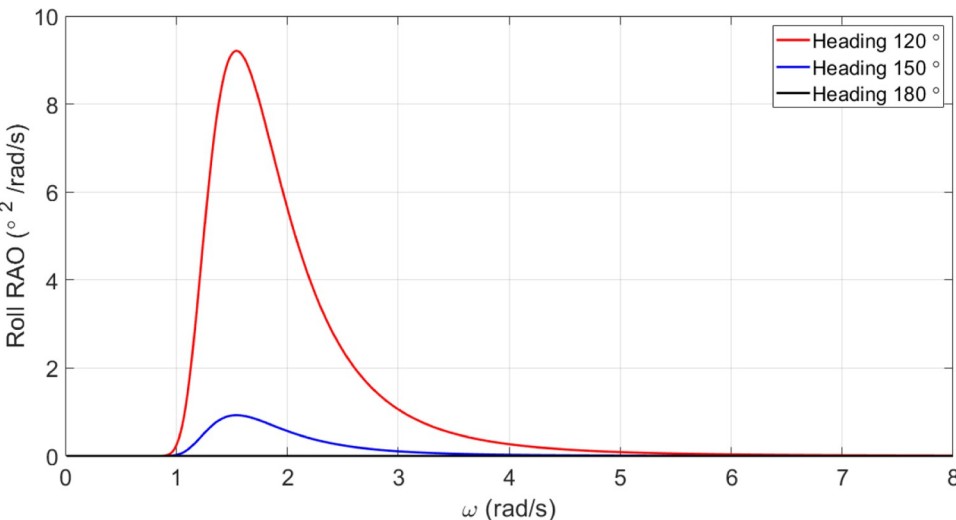

**Fig 13.** Roll motion spectrum U = 15 knots for the original model of the vessel.

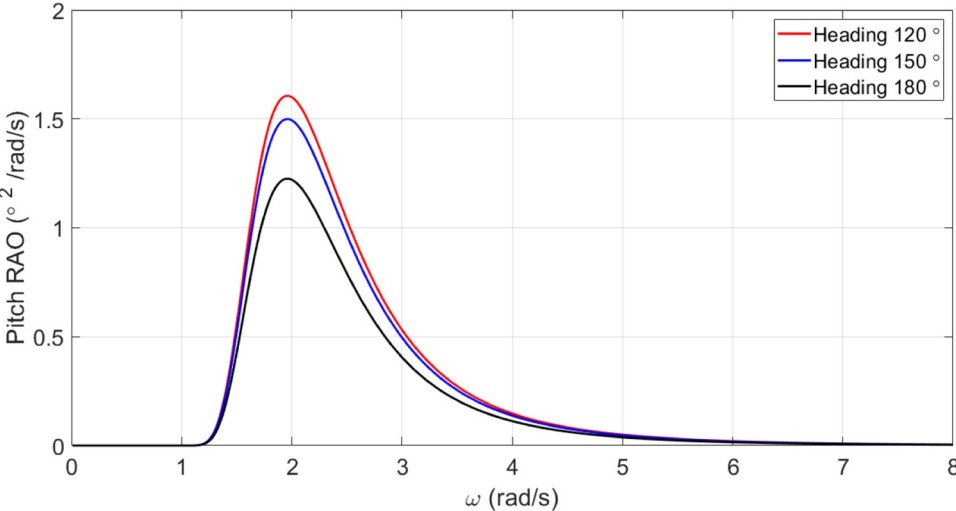

**Fig 14.**

Fig 18 illustrates that $C_M$ has a very small effect on roll motion. Increasing $C_M$, however, increases the maximum roll RAO. Conversely, decreasing $C_M$ did not have a meaningful impact on maximum roll RAO. Furthermore, an increase in $C_M$ resulted in a decrease in the frequency of occurrence of the maximum roll RAO, whereas a decrease in $C_M$ resulted in an increase in the frequency of occurrence of the maximum roll RAO.

According to Fig 19, $C_M$ had a significant impact on pitch motion. By increasing $C_M$, the maximum pitch RAO was decreased. In contrast, decreasing $C_M$ increased the maximum pitch RAO. Moreover, changes in $C_M$ had no significant effect on the frequency of occurrence of the maximum pitch RAO.

In accordance with Fig 20, $C_M$ had no significant impact on added resistance. As $C_M$ was increased, the maximum added resistance RAO was decreased. Conversely, a decrease in $C_M$ increased the maximum added resistance RAO. Also, $C_M$ changes did not significantly impact the frequency of occurrence of the maximum added resistance RAO.

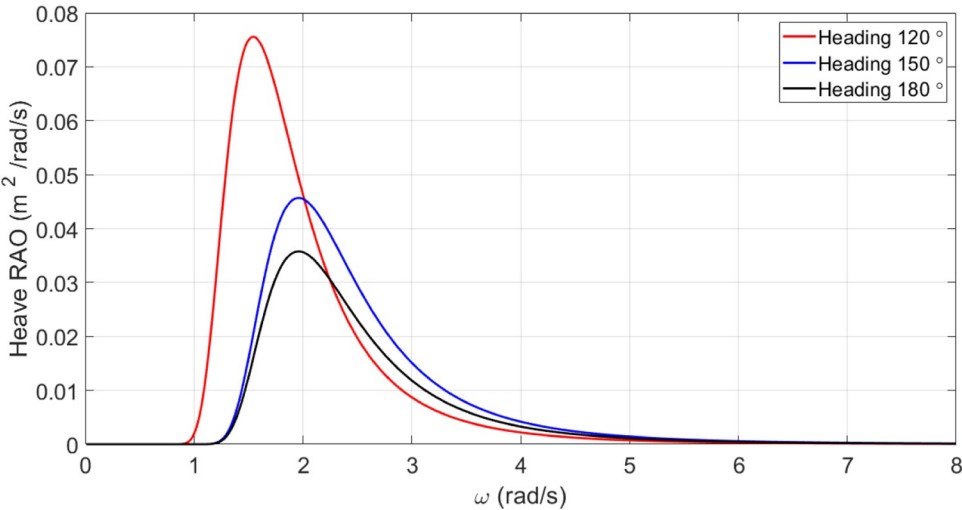

**Fig 15.** Heave motion spectrum for U = 15 knots for the original model of the vessel.

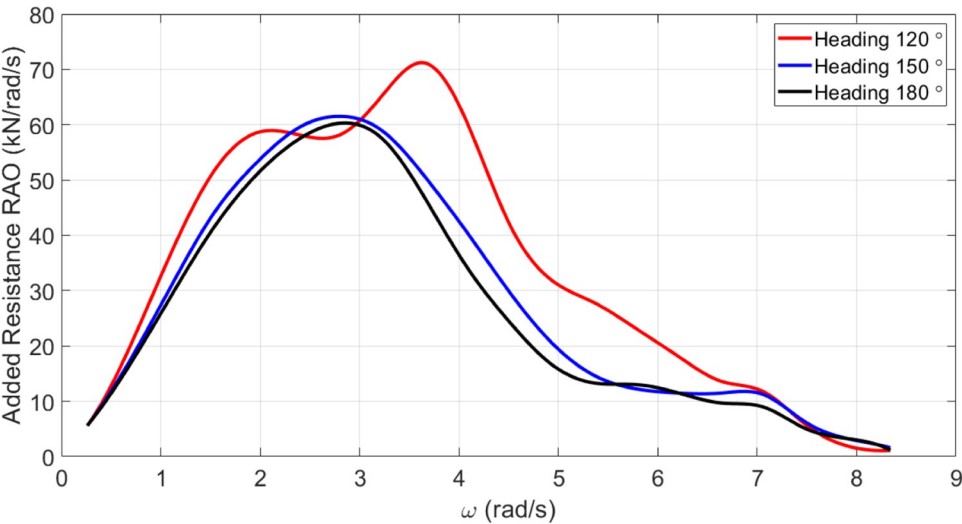

**Fig 16.** Added resistance spectrum for U = 15 knots for the original model of the vessel.

Fig 21 illustrates that $C_M$ had a significant impact on MSI. There was a direct correlation between $C_M$ changes and the maximum MSI RAO. Furthermore, the frequency of occurrence of the maximum added resistance RAO was not significantly affected by $C_M$ changes.

## 4.2. Spectrum of effective seakeeping parameters for $C_B$ coefficient

The RAO diagrams for heave, pitch, roll, added resistance, and MSI were calculated for headings of 150 degrees in the single-parameter $C_B$ spectrum at a velocity of 35 knots. Also included

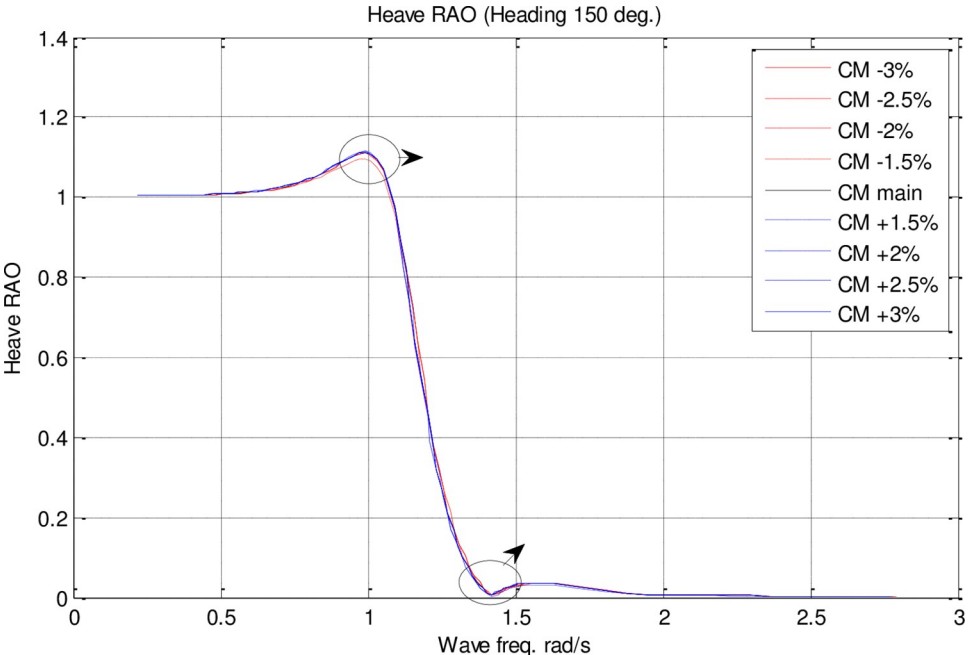

**Fig 17.** Effect of $C_M$ on the heave RAO for U=35 knots at the heading of 150˚.

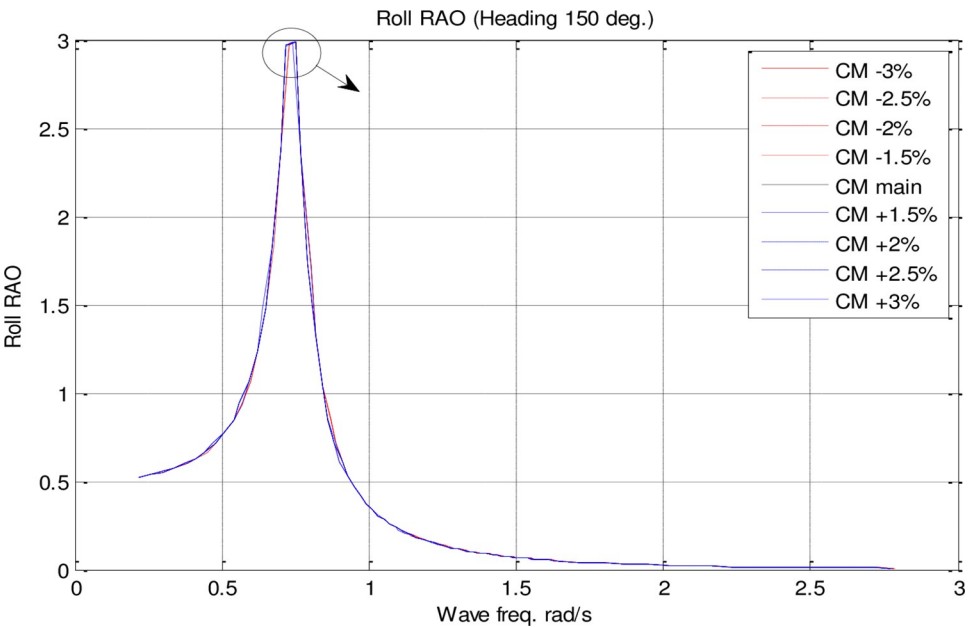

**Fig 18.** Effect of $C_M$ on the roll RAO for U=35 knots at the heading of 150˚.

are RMS diagrams for heave, pitch, and roll motions on the directional diagrams. The black diagram in these figures represents the main hull, whereas the other colors indicate the geometrical optimization models.

Fig 22 shows the heave RAO as a function of changing $C_B$ coefficients at different wave frequencies. The effect of $C_B$ on heave RAO was not significant. The maximum heave RAO, however, was increased by increasing the $C_B$. Decreased $C_B$, on the other hand, resulted in a

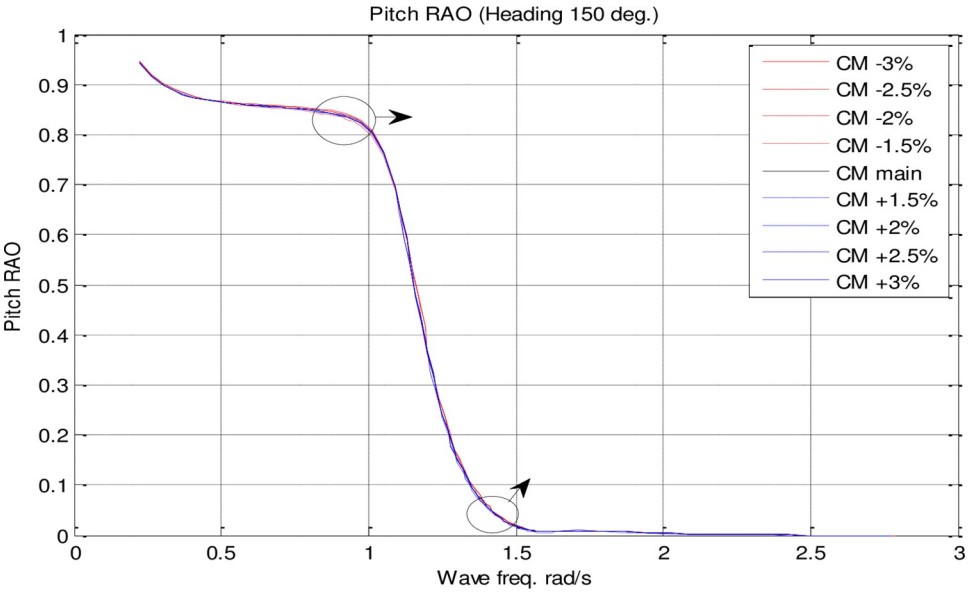

**Fig 19.** Effect of $C_M$ on the pitch RAO for U=35 knots at the heading of 150˚.

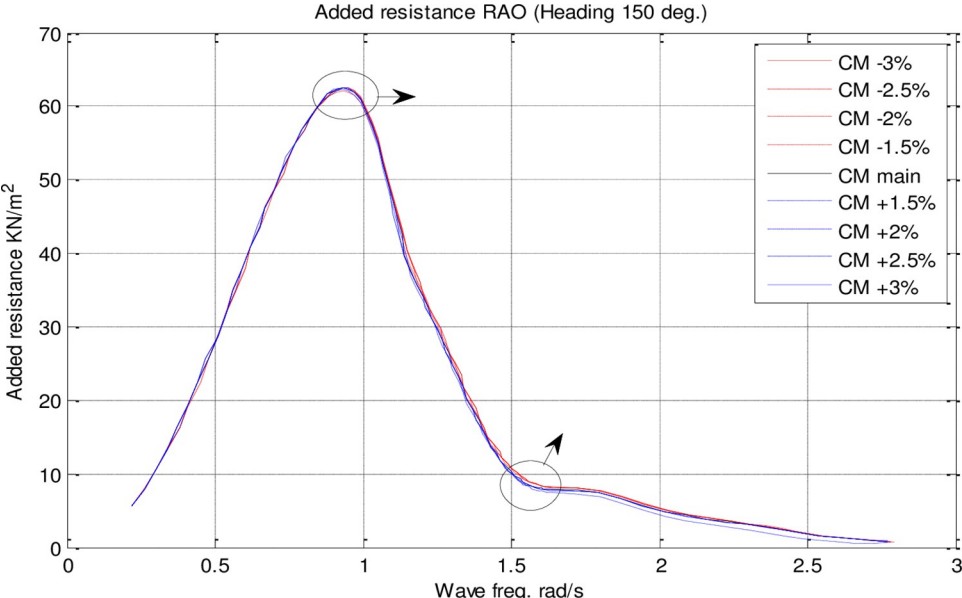

**Fig 20.** Effect of $C_M$ on the added resistance RAO for U=35 knots at the heading of 150°.

decrease in maximum heave RAO. Moreover, changes in $C_B$ did not significantly affect the frequency of occurrence of the maximum heave RAO.

Fig 23 illustrates how $C_B$ has a very small influence on roll motion. When $C_B$ is increased, however, the maximum roll RAO is increased. On the other hand, decreasing $C_B$ did not have a significant impact on maximum roll RAO. Additionally, an increase in $C_B$ decreased the frequency of occurrence of the maximum roll RAO, while a decrease in $C_B$ increased the frequency of occurrence of the maximum roll RAO.

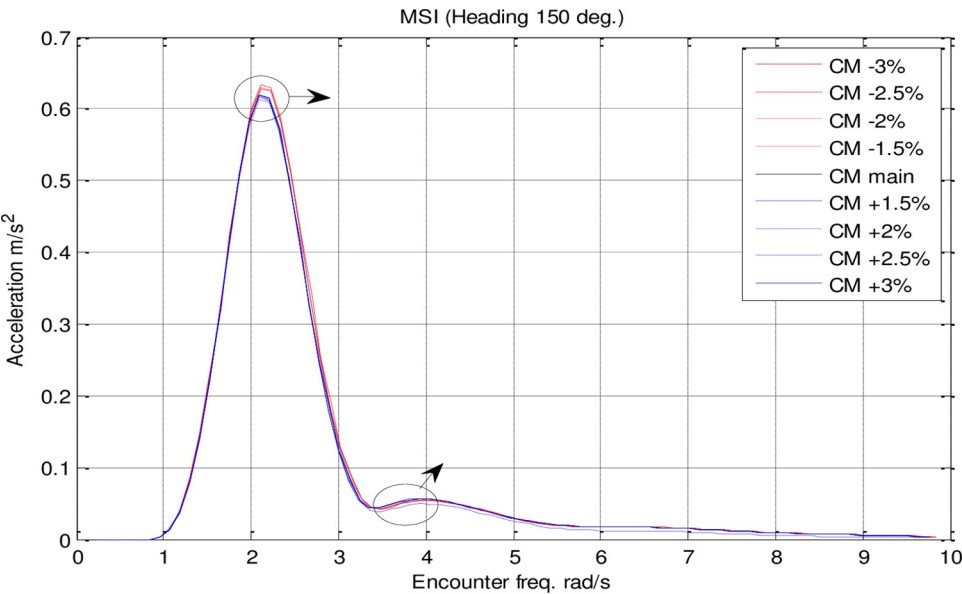

**Fig 21.** Effect of $C_M$ on the MSI acceleration RAO for U=35 knots at the heading of 150°.

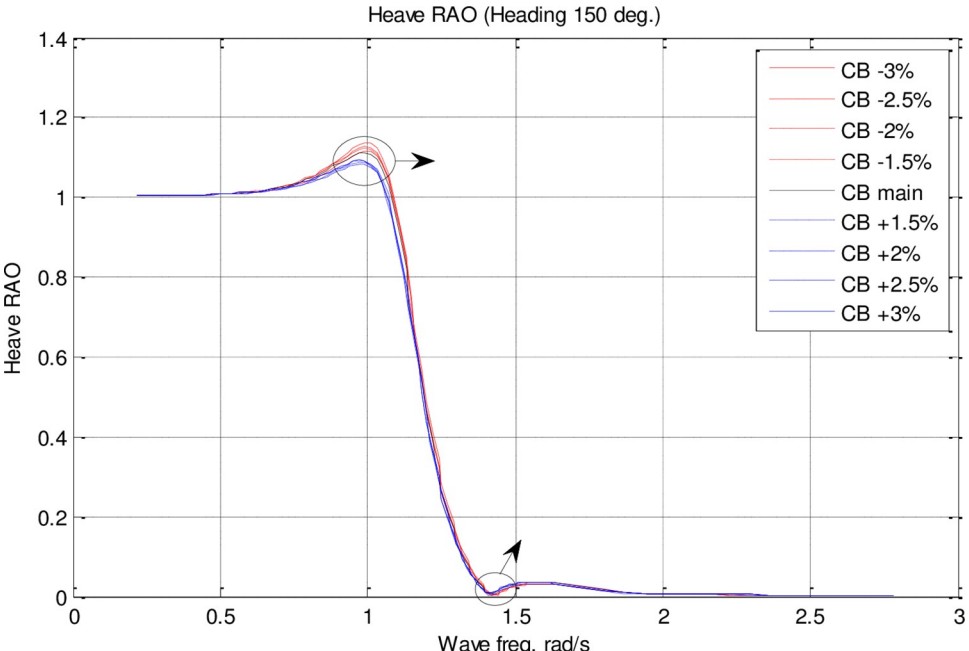

**Fig 22.** Effect of $C_B$ on the heave RAO for U=35 knots at the heading of 150˚.

Fig 24 shows that $C_B$ significantly influenced pitch motion. The maximum pitch RAO was decreased by increasing $C_B$. Conversely, a decrease in $C_B$ resulted in an increase in the maximum pitch RAO. Additionally, changes in $C_B$ did not have a significant effect on the frequency of occurrence of the maximum pitch RAO.

According to Fig 25, $C_B$ had no significant impact on resistance added. Increasing $C_B$ resulted in a decrease in maximum added resistance RAO. The maximum added resistance

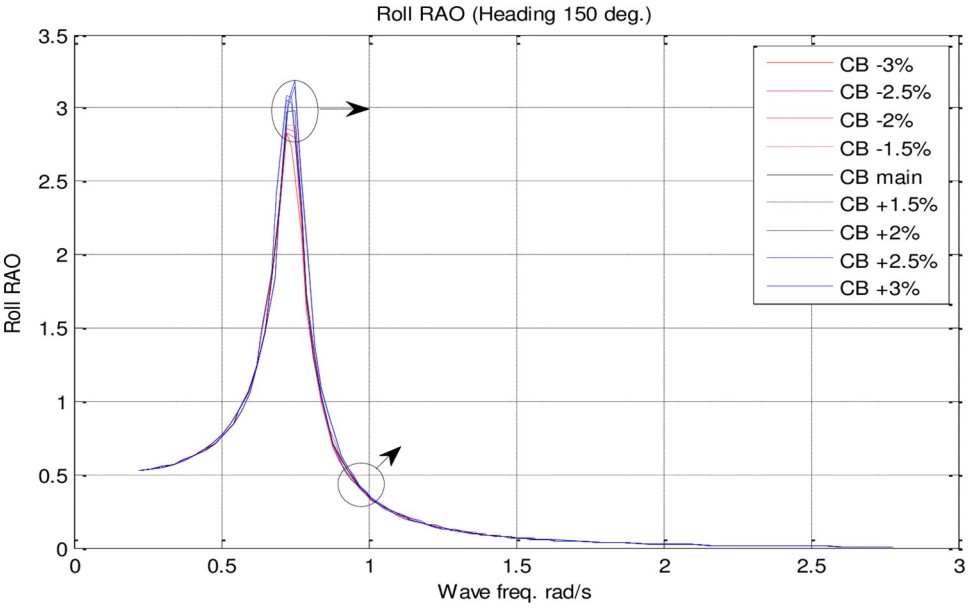

**Fig 23.** Effect of $C_B$ on the roll RAO for U=35 knots at the heading of 150˚.

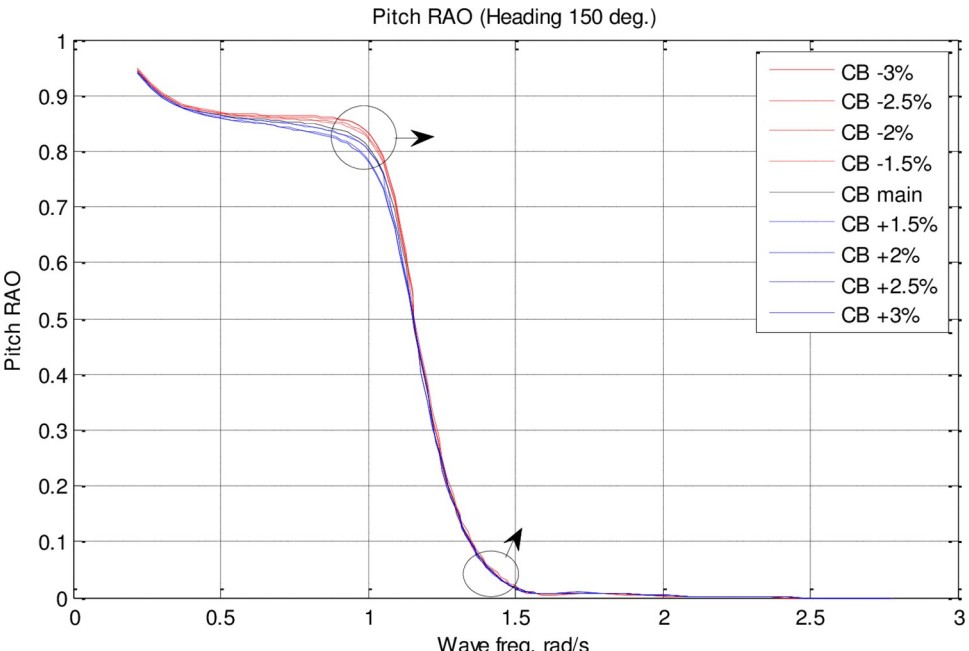

**Fig 24.** Effect of $C_B$ on the pitch RAO for U=35 knots at the heading of 150˚.

RAO increased with a decrease in $C_B$, on the other hand. Furthermore, there was no significant impact of $C_B$ changes on the frequency of occurrence of maximum added resistance RAO.

A significant impact of $C_B$ on MSI can be seen in Fig 26. It was found that $C_B$ changes had a direct correlation with the maximum MSI RAO. $C_B$ changes did not significantly affect the frequency of occurrence of the maximum added resistance RAO.

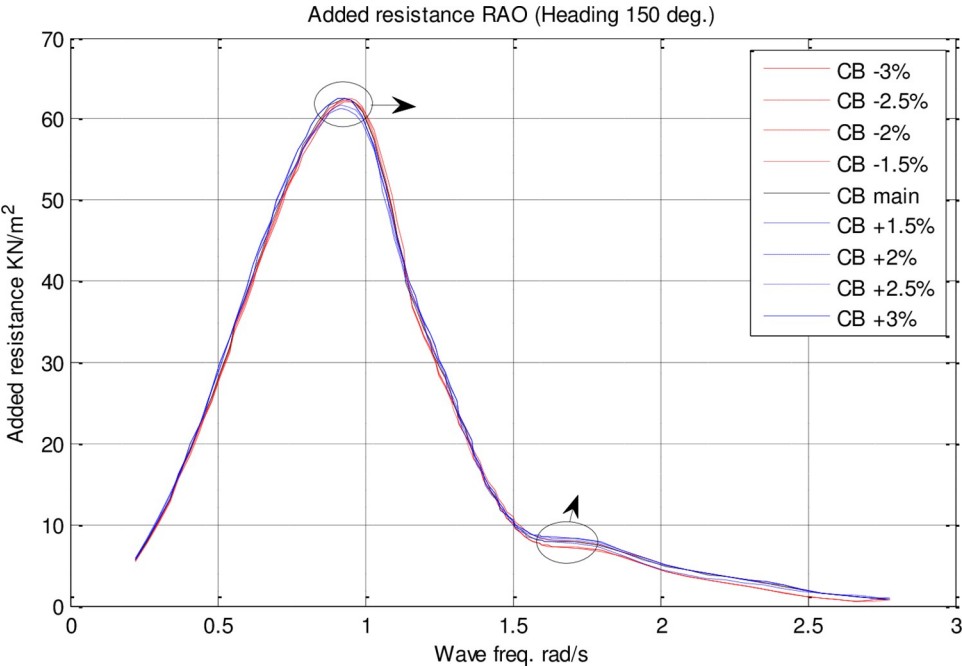

**Fig 25.** Effect of $C_B$ on the added resistance RAO for U=35 knots at the heading of 150˚.

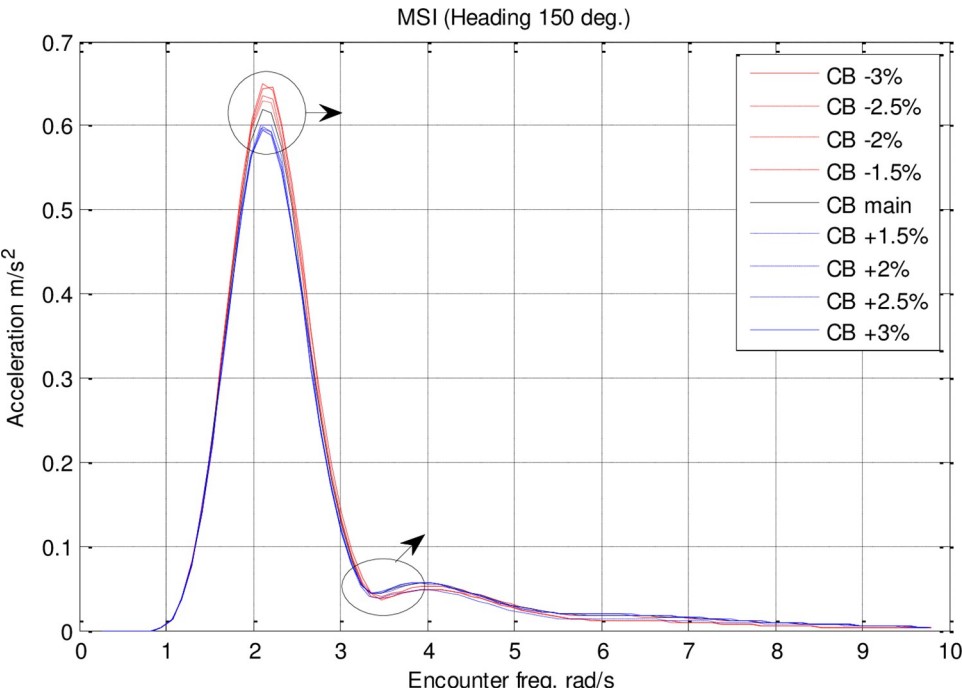

**Fig 26.** Effect of $C_B$ on the MSI RAO for U=35 knots at the heading of 150˚.

## 4.3. The final optimal solution of the geometric parameters of the vessel

In this section, a final optimal prioritized table is given for each parameter, which are numbered in order of superiority of the answer. In this way, the best answer of the first rank will be created, and this importance will decrease for the answers of lower ranks. Tables 5 and 6 show the final optimal values for midship and block coefficients, respectively.

**Table 5. Final optimum values for $C_M$.**

| Rank | $C_M$ | $\emptyset$ | $\emptyset^+$ | $\emptyset^-$ |
|---|---|---|---|---|
| 1 | 0.683095 | 0,5128 | 0,7564 | 0,2436 |
| 2 | 0.662905 | 0,4872 | 0,7436 | 0,2564 |
| 3 | 0.65281 | 0,4359 | 0,7179 | 0,2821 |
| 4 | 0.662905 | 0,3333 | 0,6282 | 0,2949 |
| 5 | 0.662905 | 0,3333 | 0,6282 | 0,2949 |
| 6 | 0.683095 | 0,0513 | 0,5256 | 0,4744 |
| 7 | 0.690579852 | -0,1026 | 0,4487 | 0,5513 |
| 8 | 0.686467634 | -0,1795 | 0,4103 | 0,5897 |
| 9 | 0.689788079 | -0,1795 | 0,4103 | 0,5897 |
| 10 | 0.688508049 | -0,3077 | 0,3462 | 0,6538 |
| 11 | 0.688660926 | -0,3077 | 0,3462 | 0,6538 |
| 12 | 0.68917637 | -0,3590 | 0,3205 | 0,6795 |
| 13 | 0.689316408 | -0,3590 | 0,3205 | 0,6795 |
| 14 | 0.689366336 | -0,3590 | 0,3205 | 0,6795 |

**Table 6. Final optimum values for $C_B$.**

| Rank | $C_B$ | $\emptyset$ | $\emptyset^+$ | $\emptyset^-$ |
|---|---|---|---|---|
| 1 | 0.458 | 0.6190 | 0.8095 | 0.1905 |
| 2 | 0.458429 | 0.5238 | 0.7619 | 0.2381 |
| 3 | 0.4740658 | 0.3571 | 0.6786 | 0.3214 |
| 4 | 0.4742765 | 0.3571 | 0.6786 | 0.3214 |
| 5 | 0.4614252 | 0.1905 | 0.5952 | 0.4048 |
| 6 | 0.471 | 0.0476 | 0.5119 | 0.4643 |
| 7 | 0.466922 | 0.0000 | 0.5000 | 0.5000 |
| 8 | 0.471 | 0.0476 | 0.4643 | 0.5119 |
| 9 | 0.458 | 0.0476 | 0.4762 | 0.5238 |
| 10 | 0.457620902 | 0.1429 | 0.4286 | 0.5714 |
| 11 | 0.4743793 | 0.1905 | 0.4048 | 0.5952 |
| 12 | 0.478812 | 0.3333 | 0.3333 | 0.6667 |
| 13 | 0.476408756 | 0.3810 | 0.3095 | 0.6905 |
| 14 | 0.478005 | 0.4524 | 0.2738 | 0.7262 |
| 15 | 0.4776814 | 0.5000 | 0.2500 | 0.7500 |

## 5. Conclusions

This study is provided optimization of ship hull form taking into consideration the block and midship coefficients ($C_M$ and $C_B$). To generate the hull form, a fuzzy model was developed. In this model, there is an indecent variation in the body lines generated by $C_M$ and $C_B$, but this does not affect the other geometric parameters. An index of seakeeping performance (SPI) measured the dynamic behavior of the vessel. $C_B$ and $C_M$ were optimized using multi-objective optimization. These are the most important findings of this study:

- The results reveal $C_B$ and $C_M$ have a significant impact on pitch and MSI which has not been shown before. The pitch motion can be uncomfortable for passengers and crew and can also affect the stability and performance of the vessel. Hence, decreasing this motion is of great importance in ship design.

- Roll motion was very little affected by both $C_B$ and $C_M$ coefficients. The maximum roll RAO was increased by increasing these coefficients, but decreasing $C_B$ had no meaningful impact on it. Furthermore, a direct relationship was found between changes in these coefficients and the frequency of occurrence of the maximum roll RAO.

- Heave RAO was not significantly impacted by block and midship coefficients. It was found that changes in these coefficients directly impacted the maximum heave RAO. Additionally, changes in these coefficients had no significant impact on the frequency of maximum heave RAO.

- Both coefficients considered had a significant impact on pitch motion. The change in these coefficients was found to have an adverse relationship with the maximum pitch RAO. There was, however, no significant effect of changes in these coefficients on the frequency of occurrence of the maximum pitch RAO.

- The coefficients for the block and midship did not have a significant impact on added resistance. This coefficient was found to have an adverse relationship with the maximum added resistance RAO. In addition, these coefficients had no significant impact on the frequency of occurrence of the maximum added resistance RAO.

- It was found that both coefficients had a significant impact on the MSI. A direct correlation was found between coefficient changes and the maximum MSI RAO. It was also observed that the frequency of occurrence of the maximum added resistance RAO was not significantly affected by coefficient changes. Decreasing the MSI improved the conformability of the crew onboard the vessel.

- Based on the mathematical procedure and performed optimization, the optimum values for block and midship coefficients were evaluated.

## Author Contributions

**Conceptualization:** Mohsen Khosravi Babadi, Hassan Ghassemi.

**Methodology:** Mohsen Khosravi Babadi, Hassan Ghassemi.

**Supervision:** Hassan Ghassemi.

**Validation:** Mohsen Khosravi Babadi, Hassan Ghassemi.

**Visualization:** Mohsen Khosravi Babadi, Hassan Ghassemi.

**Writing – original draft:** Mohsen Khosravi Babadi, Hassan Ghassemi.

**Writing – review & editing:** Mohsen Khosravi Babadi, Hassan Ghassemi.

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
