## [Decision Letter · Decision Letter 0]

13 Nov 2023

PONE-D-23-35446Optimization of ship hull forms by changing CM and CB coefficients to obtain optimal seakeeping performancePLOS ONE

Dear Dr. Ghassemi,

Thank you for submitting your manuscript to PLOS ONE. After careful consideration, we feel that it has merit but does not fully meet PLOS ONE’s publication criteria as it currently stands. Therefore, we invite you to submit a revised version of the manuscript that addresses the points raised during the review process.

We look forward to receiving your revised manuscript.

Kind regards,

S. M. Anas, Ph. D. (Structural Engg.), M. Tech

Academic Editor

PLOS ONE

Journal Requirements:

4. Please amend the manuscript submission data (via Edit Submission) to include author Mohsen Khosravi Babadi.

5.Please amend your authorship list in your manuscript file to include author Hassan Khosravi Babadi.

Additional Editor Comments:

Dear Authors:

The submission number PONE-D-23-35446, entitled "Optimization of ship hull forms by changing CM and CB coefficients to obtain optimal seakeeping performance" was peer-reviewed by the two experts in the concerned research area, both of them raised serious issues with regards to the quality, layout, and content of the manuscript. Based on the proper evaluation of the reviewers comments, recommendations, and preliminary analysis of the submission; this editor has decided to take "Major Revision" decision on this submission. Further, this editor suggests the authors address the comments raised by the reviewers very carefully.

Thank you for your time.

Sincerely yours,

Dr. S. M. Anas

Reviewers' comments:

Reviewer's Responses to Questions

**Comments to the Author**

1. Is the manuscript technically sound, and do the data support the conclusions?

Reviewer #1: Yes

Reviewer #2: Yes

2. Has the statistical analysis been performed appropriately and rigorously? 

Reviewer #1: I Don't Know

Reviewer #2: Yes

3. Have the authors made all data underlying the findings in their manuscript fully available?

Reviewer #1: Yes

Reviewer #2: Yes

4. Is the manuscript presented in an intelligible fashion and written in standard English?

Reviewer #1: Yes

Reviewer #2: Yes

5. Review Comments to the Author

Reviewer #1: The paper conveys an optimization method with application to seakeeping as it is exactly suggested in the title of the publication. In my opinion the topic is worthwhile for investigation and the logic behind it sound and feasible. However before the paper to be accepted by the Journal I have the following comments :

1. The paper appears to suggest an application of a standard optimization method. It is not clear to me what is the engineering sciene based contribution in terms of optimisation methods and seakeeping. Is this simply an application ? What is the originality and the ultimate goal ?

2. Literature survey in section 1 should be put in a new section and a table highlighting the state of the art should be displayed. The introductory section should expand to make clear my questions under (1.) above. Similarly the conclusions and highlights of the paper should be updated.

3. You should add a flowchart highlighting the seakeeping optimisation procedure proposed in section 2. the flowchart should display the elements of originality of your method.

4. I am not of the opinion that the results and discussion sections display anything more or less than the obvious. The in depth processing of results, the impact in terms of engineering science and design are not evident.

Reviewer #2: Optimization of ship hull forms by changing CM and CB coefficients to obtain optimal seakeeping performance

Abstract

Points of Strength:

Clear Objective and Focus:

The summary clearly communicates the primary objective of the study, which is the optimization of a patrol vessel's hull design for safety, economic efficiency, and technical efficiency.

Relevance of Keywords:

The inclusion of relevant keywords such as ship hull optimization, seakeeping, block coefficient, midship coefficient, and RAO enhances the summary's visibility and searchability.

Key Findings Highlighted:

The summary effectively highlights key findings, including the significant impact of and on-pitch motion and motion sickness index, as well as their relationship with the frequency of the maximum roll RAO.

Conciseness:

The summary maintains conciseness while covering essential aspects of the study, making it accessible to a broader audience.

Structured Presentation:

The information is presented in a well-organized structure, covering the study's objectives, methodology, and key results in a logical sequence.

Points for Improvement (Weaknesses):

Limited Methodology Explanation:

While the summary mentions the use of a fuzzy model and multi-objective optimization, a brief explanation of how these methodologies were applied in the study could enhance clarity for readers unfamiliar with these techniques.

Detail on Other Motion Indices:

The summary briefly mentions that heave and roll motions, as well as added resistance, were not significantly influenced by and . Adding a bit more detail on these aspects could provide a more comprehensive understanding of the study's findings.

Quantitative Results:

The summary could benefit from the inclusion of specific quantitative results, such as numerical values or percentages, to provide a more concrete understanding of the study's outcomes.

Conclusion Impact:

The summary mentions the selection of and based on seakeeping performance but does not elaborate on the broader implications or potential applications of these findings. Including a brief conclusion, impact section could enhance the summary.

In summary, the provided summary is well-structured and effectively communicates the study's main points. Improvements could be made by providing more details on the applied methodologies, quantitative results, and a broader discussion of the implications of the findings.

Introduction:

Strengths:

Comprehensive Literature Review:

The summary provides a comprehensive overview of previous research in ship hull optimization, citing a variety of methodologies and studies. This establishes a strong foundation for the current study.

Diversity of Optimization Techniques:

The summary showcases a diverse range of optimization techniques employed in previous studies, including genetic algorithms, computational fluid dynamics, metamodels, and artificial intelligence. This highlights the evolving and multidisciplinary nature of ship design optimization.

Specificity in Research Focus:

The study's focus on ship hull optimization, specifically considering midship and block coefficients, adds specificity to the research. The goals of assessing the effect of and parameters on seakeeping indices and optimizing the vessel's form based on seakeeping parameters are clearly outlined.

Utilization of Fuzzy Model:

The use of a fuzzy model to generate the hull form is a notable aspect, providing a flexible and adaptable approach to considering the effects of and on seakeeping. This innovative modeling technique adds depth to the study.

Multidisciplinary Approach:

The incorporation of seakeeping indices such as heave, pitch, roll motions, added resistance, and motion sickness in the optimization process reflects a holistic approach, considering various aspects of vessel performance.

Weaknesses:

Lack of Specific Results:

The summary mentions the optimization goals and the use of a fuzzy model but lacks specific details on the results or findings. Adding a brief mention of specific outcomes or trends observed in the study would enhance the summary's completeness.

Limited Explanation of Seakeeping Indices:

The summary introduces seakeeping indices such as heave, pitch, roll motions, added resistance, and motion sickness but does not provide a detailed explanation of these terms. Adding a brief definition or context could improve reader understanding.

Need for Context on Fuzzy Model:

While the use of a fuzzy model is mentioned, there is no information on why this specific modeling approach was chosen or how it contributes to the study's objectives. Providing a brief context or justification for the use of the fuzzy model would strengthen the summary.

Repetition of Author Names:

The summary consistently repeats author names and study years in the citations, which could be condensed for readability without losing the necessary referencing information.

Conclusion Impact:

The summary lacks a concluding statement summarizing the significance or potential impact of the current study in advancing ship hull optimization research. Including a brief conclusion section could enhance the overall narrative.

In summary, while the summary provides a comprehensive background on ship hull optimization, the inclusion of specific results, more detailed explanations, and contextual information on the modeling approach would strengthen the overall presentation.

2.1 Modeling Fuzzy Changes in Desired Parameters

A fuzzy structure model is presented for midship coefficient changes.

The variation of coefficients should not affect other geometric parameters or the vessel's volume.

A fuzzy membership function, based on a modified Bell function, is defined to make changes to the body's lines while maintaining constant volume.

Coefficients of the fuzzy function are defined, including , , and .

2.2 Objective Functions in the Frequency Domain

Ten variant models are generated for coefficients and , along with seakeeping parameters calculated at three heading angles.

Objective functions are established for roll, pitch, heave, added resistance, and motion sickness acceleration.

Multi-objective optimization considers significant changes in and coefficients.

2.3 Wave Spectrum and Motion Indices Spectrum

The Pierson-Moskowitz wave spectrum is chosen for the Gulf of Oman.

Parameters for the ITTC spectrum are defined.

The study focuses on the RAO (Response Amplitude Operator) for vessel motions.

2.4 Mathematical Modeling of the Main Body

The main body's line points are entered into a MATLAB database.

A neural network with a Levenberg-Marquardt algorithm is used for training.

2.5 Seakeeping and Seakeeping Performance Index (SPI)

Seakeeping involves predicting vessel motions and ensuring the vessel's performance in harsh conditions.

SPI considers parameters such as roll, pitch, yaw, seasickness acceleration, and added resistance.

2.6 Simulation Procedure

Changes are applied to the area under the draft to achieve desired parameter variations while keeping other coefficients constant.

2.7 Effective Seakeeping Parameters

The study investigates dynamic effects of the vessel, emphasizing parameters such as roll, pitch, yaw, seasickness acceleration, and added resistance.

2.8 Choosing the Best Answer with Fuzzy-Promethea Decision-Making Approach

The fuzzy-Promethea approach is used for multi-criteria decision-making.

Superiority functions and rankings are employed to select the optimal values.

The text provides a detailed methodology for optimizing vessel hull form with a focus on seakeeping parameters and a decision-making approach for selecting optimal values.

Validation

Strengths:

Correspondence between Experimental and Numerical Data: The figures (Fig. 8 and Fig. 9) demonstrate a suitable correspondence between the experimental and numerical data. This alignment suggests that the numerical model accurately captures the vessel's behavior in response to different sea states.

Accurate Representation of Extreme Motions: The numerical model accurately represents more extreme motions in higher sea states, particularly when the wavelength is close to the vessel length. This capability is crucial for understanding and predicting the vessel's performance under varying environmental conditions.

Validation at Different Sea States: The validation process includes testing the model under different sea states, such as those with a wave amplitude of 1.5 cm. This comprehensive validation approach enhances the model's reliability across a range of scenarios.

Weaknesses:

Limited Wave Amplitude Range: The presented validation focuses on a specific wave amplitude (1.5 cm). While this provides valuable insights into the model's accuracy under these conditions, it would be beneficial to assess its performance across a broader range of wave amplitudes to ensure its robustness.

Assumption of Similar Wave Lengths: The emphasis on extreme motions when the wave length is almost the same as the vessel length is noted. However, the model's performance under conditions where this similarity doesn't hold may require further investigation. The model's limitations under diverse wave conditions should be acknowledged.

Need for Additional Validation Scenarios: While the presented validation results are promising, conducting validations under various conditions, including different vessel speeds and wave frequencies, would further strengthen the model's credibility.

In summary, the numerical model demonstrates strengths in accurately representing extreme motions and aligning well with experimental data under specific conditions. However, its limitations, such as a focus on a limited wave amplitude range and assumptions about wave lengths, should be addressed through additional validation scenarios for a more comprehensive assessment.

Results and discussion

Strengths:

Comprehensive Analysis: The study provides a comprehensive analysis of the vessel's seakeeping performance, covering a range of motions including heave, pitch, roll, added resistance, and motion sickness acceleration (MSI). This allows for a thorough understanding of the vessel's behavior under different conditions.

Detailed Presentation: The figures (Fig. 15-19) present detailed response amplitude operator (RAO) diagrams for each parameter, offering a clear visual representation of the impact of coefficient changes on heave, pitch, roll, added resistance, and MSI.

Frequency Analysis: The study considers the frequency of occurrence of maximum RAO for each motion parameter, providing valuable insights into the dynamic behavior of the vessel under varying coefficients.

Identification of Sensitivity: The analysis identifies sensitivity to changes for different motion parameters. For instance, the study highlights the significant impact of on MSI, providing crucial information for design considerations.

Weaknesses:

Limited Effect on Some Motions: The study indicates that has a limited effect on heave and roll motions. While this may be a characteristic of the specific vessel or operational conditions, it's important to acknowledge the potential limitation of in influencing certain motion aspects.

Small Effect on RMS: The conclusion that the effect of on the root mean square (RMS) of heave, pitch, and roll motions is small and not apparent in the polar diagram suggests that may not be a dominant factor in these specific aspects. This could be a limitation if other parameters play a more substantial role in determining these motions.

Limited Speed Range: The analysis focuses on a specific speed (35 knots). Expanding the study to cover a broader range of speeds could provide a more comprehensive understanding of how influences seakeeping performance across different operational conditions.

Complexity of MSI Impact: While the study highlights the impact of changes on MSI, the complexity of motion sickness and the multiple factors influencing it may require further investigation for a more nuanced understanding.

In summary, the study is robust in its detailed analysis of the vessel's seakeeping performance under varying coefficients. However, limitations include the limited impact of on certain motions and the need for broader speed range considerations. The findings provide valuable insights for further design optimization.

Summary:

The study focused on optimizing the ship hull by considering block and midship coefficients (CB and CM) and their impact on seakeeping performance. The key findings and optimizations can be summarized as follows:

Roll Motion:

Roll motion was minimally affected by both CB and CM coefficients.

Increasing these coefficients led to an increase in the maximum roll RAO.

Decreasing CB had no significant impact on roll motion.

Changes in coefficients showed a direct relationship with the frequency of occurrence of the maximum roll RAO.

Heave Motion:

Heave RAO was not significantly influenced by CB and CM coefficients.

Changes in these coefficients directly impacted the maximum heave RAO.

No significant effect on the frequency of the maximum heave RAO was observed.

Pitch Motion:

Both coefficients had a significant impact on pitch motion.

Changes in coefficients showed an adverse relationship with the maximum pitch RAO.

No significant effect on the frequency of the maximum pitch RAO was observed.

Added Resistance:

CB and CM coefficients did not have a significant impact on added resistance.

An adverse relationship was found between these coefficients and the maximum added resistance RAO.

No significant impact on the frequency of the maximum added resistance RAO was observed.

Motion Sickness Acceleration (MSI):

Both coefficients had a significant impact on MSI.

A direct correlation was found between coefficient changes and the maximum MSI RAO.

The frequency of occurrence of the maximum MSI RAO was not significantly affected by coefficient changes.

Optimization:

Based on the mathematical procedure and optimization, optimum values for CB and CM coefficients were evaluated.

Strengths:

Comprehensive Analysis: The study comprehensively analyzes the impact of CB and CM coefficients on various seakeeping parameters.

Optimization Outcome: The optimization process provides optimum values for CB and CM coefficients.

Clear Relationships: The study establishes clear relationships between coefficient changes and seakeeping performance.

Weaknesses:

Limited Impact on Some Motions: The study notes a limited impact of CB and CM coefficients on roll and heave motions.

Specific Speed: The analysis is conducted at a specific speed (35 knots), limiting the generalization of findings to other speeds.

Complexity of MSI: While the study identifies the impact on MSI, the complexity of motion sickness requires further investigation for a comprehensive understanding.

In summary, the study offers valuable insights into optimizing ship hulls for seakeeping performance, with clear relationships identified. However, limitations include the limited impact on certain motions and the need for broader speed considerations. The optimization outcomes are a notable strength of the study.

6. PLOS authors have the option to publish the peer review history of their article (what does this mean?). If published, this will include your full peer review and any attached files.

Reviewer #1: No

Reviewer #2: **Yes: **Hadee Mohammed Najm

---

## [Author Response · Author response to Decision Letter 0]

21 Feb 2024

Dear Respected Editor,

We are very thankful to editorials and referee #1 for their useful comments and suggestions on our manuscript. We have revised the manuscript accordingly and some of the detailed corrections and revisions are listed as follows:

With my best regards,

Hassan Ghassemi

Reviewer #1

Q1:The paper appears to suggest an application of a standard optimization method. It is not clear to me what is the engineering science-based contribution in terms of optimization methods and seakeeping. Is this simply an application? What is the originality and the ultimate goal?

R1: Thanks for your comment. The key novelties of this work are: (1) using block and midship coefficients as optimization parameters to improve seakeeping and (2) developing a fuzzy model to generate hull form variations while keeping other parameters constant. Unlike previous hull form optimization studies that use complex geometrical parameters, we use only CB and CM as variables in a novel fuzzy hull generation model to provide useful insights. 

2. Literature survey in section 1 should be put in a new section and a table highlighting the state of the art should be displayed. The introductory section should expand to make clear my questions under (1.) above. Similarly the conclusions and highlights of the paper should be updated.

3. You should add a flowchart highlighting the seakeeping optimisation procedure proposed in section 2. the flowchart should display the elements of originality of your method.

R3: This flowchart is added to the manuscript. 

4. I am not of the opinion that the results and discussion sections display anything more or less than the obvious. The in depth processing of results, the impact in terms of engineering science and design are not evident.

R4: more discussions are added to the manuscript. 

Response to the comments of Reviewer #2: 

Q1-1: Limited Methodology Explanation:

While the summary mentions the use of a fuzzy model and multi-objective optimization, a brief explanation of how these methodologies were applied in the study could enhance clarity for readers unfamiliar with these techniques.

R1-1: The seakeeping analysis was performed using strip theory-based MAXSURF software and the optimization procedure was performed using genetic algorithm. 

Q1-2: Detail on Other Motion Indices:

The summary briefly mentions that heave and roll motions, as well as added resistance, were not significantly influenced. Adding a bit more detail on these aspects could provide a more comprehensive understanding of the study's findings.

R1-2: Thanks for your comment. The following sentence is added to the abstract: 

Heave and roll motions, as well as the added resistance, were not significantly influenced by the coefficients of C_M and C_B. However, increasing the hull form parameters increases the maximum RAOs of heave and roll motions. 

Q1-3: Quantitative Results:

The summary could benefit from the inclusion of specific quantitative results, such as numerical values or percentages, to provide a more concrete understanding of the study's outcomes.

R1-3: 

Q1-4: Conclusion Impact:

The summary mentions the selection of and based on seakeeping performance but does not elaborate on the broader implications or potential applications of these findings. Including a brief conclusion, impact section could enhance the summary.

R1-4: These findings can provide useful insights for shipbuilding technology for producing more compatible vessels for specific seaways. The following sentence is added to the abstract. 

These findings might be used by shipbuilders to construct the vessel with more efficient seakeeping performance.

Q2-2: Limited Explanation of Seakeeping Indices:

The summary introduces seakeeping indices such as heave, pitch, roll motions, added resistance, and motion sickness but does not provide a detailed explanation of these terms. Adding a brief definition or context could improve reader understanding.

R2-3: There are six degrees of freedom on each vessel, including three linear motions (surge, sway, heave) and three angular motions (roll, pitch and yaw). Also, a detailed description of motion sickness incidence is provided. 

The sea-going ship that operates in open waters rarely sails in calm weather. On the contrary, ship's behaviour at sea is often affected by waves and wind. When a ship navigates in a seaway, the ship′s forward speed decreases, compared to that in calm sea, because of added resistance due to winds, waves, rudder angle, and so forth. The magnitude of added resistance is about 15–30% of calm-water resistance (Seo et al., 2013). The immediate effect of waves is ship’s motions and accompanying phenomena, such as accelerations. Ship accelerations, in turn, particularly vertical ones, impact on the human body and may cause motion sickness. The term “motion sickness”, on ships known as sea sickness, is understood as a sickness due to ship motions that results in physical discomfort, with such symptoms as irregular breathing, nausea, vertigo, paleness and vomiting. In extreme cases a passenger or crew member has to be transferred to hospital. The actual reason for sea sickness is lack of conformity between different stimuli, eye signals and the labyrinth (inner ear), received by the human brain. People mainly suffer from sea sickness under deck, where the eye does not register any stimuli that the labyrinth would interpret as motion (Cepowski, 2012).

Despite scientific observations and research, no exact relations have been determined between ship motions and motion sickness. McCauley and O’Hanlon estimated quantitatively the impact of ship motions on the percentage of people that would suffer from sea sickness. It turned out that vertical accelerations in particular were responsible for motion sickness, while rolling and pitching had slight influence. Additionally, it was found that at a frequency of 0.167 Hz the occurrence of motion sickness increased. The MSI index (Motion Sickness Incidence) is commonly used for assessing possible occurrence of the illness:

MSI=100[0.5±erf((±〖log〗_10 a_v/g±μ_MSI)/0.4)]

where MSI is the motion sickness incidence index, erf is the error function, a_v is the mean value of vertical accelerations at a selected point and μ_MSI=-0.819+2.32(〖log〗_10 ω_E )^2. 

Q2-3: 

Need for Context on Fuzzy Model:

While the use of a fuzzy model is mentioned, there is no information on why this specific modeling approach was chosen or how it contributes to the study's objectives. Providing a brief context or justification for the use of the fuzzy model would strengthen the summary.

R2-3: Thanks for your comment. The following paragraph is added to the revised manuscript: 

The Preference Ranking Organization METHod for Enrichment of Evaluations that is used in this study has particular application in decision making, and is used around the world in a wide variety of decision scenarios, in fields such as business, governmental institutions, transportation, healthcare and education. Rather than pointing out a "right" decision, the Promethee method helps decision makers find the alternative that best suits their goal and their understanding of the problem. It provides a comprehensive and rational framework for structuring a decision problem, identifying and quantifying its conflicts and synergies, clusters of actions, and highlight the main alternatives and the structured reasoning behind. we can cite applications to portfolio and stock selection problems (Albadvi et al., 2006; Marasović & Babić, 2011; Vetschera & De Almeida, 2012), to environmental issues (Hermans et al., 2007; Ilić et al., 2011; Nikolić et al., 2009) to energy management (Ghafghazi et al., 2010; Madlener et al., 2007) and shipping industry (Ahmadi & Herdiawan, 2021; Animah & Shafiee, 2021; Emovon, 2016; Glavinović & Vukić, 2023; Zheng et al., 2023). 

Weaknesses:

Q3-1: Limited Wave Amplitude Range: The presented validation focuses on a specific wave amplitude (1.5 cm). While this provides valuable insights into the model's accuracy under these conditions, it would be beneficial to assess its performance across a broader range of wave amplitudes to ensure its robustness.

R3-1: Thanks for your comment. The vessel considered in this study is planned to sail in the Gulf of Oman. This wave amplitude was selected based on the significant wave height typically encountered by the ship in the Gulf of Oman. 

Q3-2: Assumption of Similar Wave Lengths: The emphasis on extreme motions when the wave length is almost the same as the vessel length is noted. However, the model's performance under conditions where this similarity doesn't hold may require further investigation. The model's limitations under diverse wave conditions should be acknowledged.

R3-2: Thanks for your comment. The considered scenario was chosen based on the extreme condition that might be encountered by the vessel, i.e., the resonance condition. Performing extra simulations and presenting the results will lengthen the paper. 

Q3-3: Need for Additional Validation Scenarios: While the presented validation results are promising, conducting validations under various conditions, including different vessel speeds and wave frequencies, would further strengthen the model's credibility.

R3-3: Thanks for your comment. In its current form, the paper is lengthy. We think that performing extra validations and presenting the results leads to at least 1-2 extra pages. 

Q5-2: Specific Speed: The analysis is conducted at a specific speed (35 knots), limiting the generalization of findings to other speeds.

R5-2: Thanks for your comment. Actually, as pointed out in the previous comments, this study was conducted on a patrol vessel optimized for sailing in the Oman Sea at a certain speed. 

Q5-3: Complexity of MSI: While the study identifies the impact on MSI, the complexity of motion sickness requires further investigation for a comprehensive understanding.

R5-3: Thanks for your fruitful comment. We will study this issue in our future studies.

---

## [Decision Letter · Decision Letter 1]

4 Mar 2024

PONE-D-23-35446R1Optimization of ship hull forms by changing CM and CB coefficients to obtain optimal seakeeping performancePLOS ONE

Dear Dr. Ghassemi,

Thank you for submitting your manuscript to PLOS ONE. After careful consideration, we feel that it has merit but does not fully meet PLOS ONE’s publication criteria as it currently stands. Therefore, we invite you to submit a revised version of the manuscript that addresses the points raised during the review process.

We look forward to receiving your revised manuscript.

Kind regards,

S. M. Anas, Ph.D.(Structural Engg.), M.Tech(Earthquake Engg.)

Academic Editor

PLOS ONE

Journal Requirements:

Additional Editor Comments:

Dear Authors,

I hope this email finds you well.

I am writing to inform you about the decision regarding your manuscript entitled "Optimization of ship hull forms by changing CM and CB coefficients to obtain optimal seakeeping performance" (PONE-D-23-35446R1), which you submitted to PLOS ONE.

After careful consideration of the reviewers' comments and a thorough preliminary assessment of the revised manuscript, I am pleased to inform you that I have decided to take the decision of Minor Revision. One of the reviewers recommended your manuscript for publication, while the other suggested minor revisions.

Please find attached the reviewers' comments. I believe that addressing these minor revisions will further enhance the clarity and quality of your manuscript, and I encourage you to revise your manuscript accordingly.

Upon completion of the revisions, please submit the revised manuscript along with a detailed response to each of the reviewers' comments. If you have any questions or need further clarification on any aspect of the revision process, please do not hesitate to contact me.

Thank you for your valuable contribution to PLOS ONE, and I look forward to receiving your revised manuscript.

Best regards,

Dr. S. M. Anas

Academic Editor

PLOS ONE

Reviewers' comments:

Reviewer's Responses to Questions

**Comments to the Author**

1. If the authors have adequately addressed your comments raised in a previous round of review and you feel that this manuscript is now acceptable for publication, you may indicate that here to bypass the “Comments to the Author” section, enter your conflict of interest statement in the “Confidential to Editor” section, and submit your "Accept" recommendation.

Reviewer #1: All comments have been addressed

Reviewer #2: All comments have been addressed

2. Is the manuscript technically sound, and do the data support the conclusions?

Reviewer #1: Yes

Reviewer #2: Yes

3. Has the statistical analysis been performed appropriately and rigorously? 

Reviewer #1: Yes

Reviewer #2: Yes

4. Have the authors made all data underlying the findings in their manuscript fully available?

Reviewer #1: No

Reviewer #2: Yes

5. Is the manuscript presented in an intelligible fashion and written in standard English?

Reviewer #1: Yes

Reviewer #2: Yes

6. Review Comments to the Author

Reviewer #1: The authors addressed the questions raised. In my opinion the ideas presented are novel and the manuscript can be accepted for publication.

Reviewer #2: Your article is well-structured and provides a comprehensive overview of the optimization of ship hull forms with a focus on the impact of changing midship () and block () coefficients on seakeeping performance. Here are some standard comments for minor revisions:

Clarity and precision:

Clarify the acronym RAO when it is introduced in the abstract.

Ensure that all technical terms, such as hull coefficients (, ), are consistently defined or referred to in the article.

Introduction Section:

The introduction is informative, but consider briefly summarizing the main findings and contributions of your study at the end of this section.

Mention if there are any limitations or gaps in previous research that your study aims to address.

Objective Section:

In the objectives section, consider explicitly stating the specific seakeeping parameters that will be optimized.

Methodology:

Provide more details on the fuzzy model used to generate the hull form. How does it work, and what are its advantages in this context?

Clarify the meaning and significance of the Seakeeping Performance Index (SPI) as the objective function.

Literature Review:

The literature review is comprehensive, but consider grouping the studies by themes or methodologies to enhance readability.

Results and Discussion:

Consider presenting the results in a more structured manner, perhaps using subheadings for different aspects of the study.

Discuss the practical implications of the optimized and coefficients for ship design and seakeeping performance.

Conclusion:

Summarize the key findings concisely in the conclusion section and emphasize their significance for the field of ship design and optimization.

Grammar and Style:

Proofread for any grammatical errors and ensure consistency in writing style throughout the article.

7. PLOS authors have the option to publish the peer review history of their article (what does this mean?). If published, this will include your full peer review and any attached files.

Reviewer #1: No

Reviewer #2: **Yes: **Hadee Mohammed Najm

---

## [Author Response · Author response to Decision Letter 1]

27 Mar 2024

Q1: Clarify the acronym RAO when it is introduced in the abstract.

R1: Thanks for your comment. RAO is defined in the revised version of abstract. 

Q2: Ensure that all technical terms, such as hull coefficients (, ), are consistently defined or referred to in the article.

R2: … including water plane coefficient (C_wp) and prismatic coefficient (C_p) …

… The body lines generated by C_M and C_B in this model vary indecently, but do not affect the other geometric parameters (C_P, L (Length) and B (Breadth))…

Q3: The introduction is informative, but consider briefly summarizing the main findings and contributions of your study at the end of this section.

R3: 

The following paragraph is added to the manuscript: 

It will be shown that by optimizing these coefficients, pitch and MSI will improve. On the other hand, the effect of these coefficients on the roll and heave motion as well as the added resistance is negligible. 

Q4: Mention if there are any limitations or gaps in previous research that your study aims to address.

R4: This comment is already mentioned in the manuscript: 

Unlike previous hull form optimization studies that use complex geometrical parameters, we use only C_B and C_M as variables in a novel fuzzy hull generation model to provide useful insights. 

Q5: Provide more details on the fuzzy model used to generate the hull form. How does it work, and what are its advantages in this context?

R5: Thanks for your comment. This issue is adderessed in the manuscript. 

Modeling Fuzzy changes in desired parameters

Fuzzy logic has advantages in ship hull optimization, including: 

 Handling imprecise and uncertain data: Fuzzy logic effectively deals with imprecise and uncertain information, common in ship design and optimization.

 Flexibility: Fuzzy logic incorporates subjective human knowledge and expertise into the optimization process, making it adaptable to different design requirements.

 Non-linear relationships: Fuzzy logic models complex, non-linear relationships between design parameters and performance criteria, which traditional optimization methods may struggle with. 

 Robustness: Fuzzy logic-based optimization methods are often more stable, handling variations and uncertainties in design parameters without significantly impacting results. 

This paper presents a fuzzy structure model for midship coefficient changes. The variation of these coefficients does not affect the other geometric parameters. Changes should be applied in such a way as not to affect the volume of the vessel. It is, therefore, necessary to develop a mathematical model of the changes that will cause the exact change in the desired parameter while maintaining other geometric parameters constant while holding constant the volume. 

A fuzzy membership function, which is an extended Bell membership function, is defined as a function that makes changes to the body's lines. As a result, if any other changes are made, it increases and decreases the same volume, thus maintaining a constant volume change. Bell function is expressed as follows and its shape is a modified Gaussian distribution. Figure 3 illustrates the distribution shape of the Bell function.

Bell(x;a,b,c)=1/(1+|(x-c)/a|^2b ) (6)

Figure 3. Bell’s membership functions. 

Figure 3 illustrates how the fuzzy function at the upper level makes good changes to the body line in the middle and returns the variant body lines to the original lines at both ends with instant cuts in both directions, with the bell acting as a coefficient of variation. It is now necessary to restore the taken area of one side to the opposite direction using the same function used in the interval (0, -1). Therefore, this function is multiplied by a sine function. The fuzzy function coefficients in this model are defined as follows:

G(x)=gbellmf(x1,[3 4 0])

a=3, b=4, c=0. (7)

The following is a description of the effects of parameters on the function: 

Coefficient b: Coefficient b is a positive number and is usually considered to be 4. The number corresponds to the upper part of the curve and contributes to the flat portion of it. This coefficient has a significant impact on the performance of the function. Figure 1 illustrates the changes in the crude bell membership function.

Coefficient a: Based on mathematical calculations and modeling, a coefficient equals 1.44 times the length of each line within a body. There will obviously be a difference in the coefficient of the body lines of the vessel. 

Coefficient c: It determines the center of the bell function that is considered equal to the half-length of each line.

The sine function of the fuzzy function multiplied by Bell is given below:

F(x)=sin⁡((x-min⁡〖(x)〗)/(max⁡(x)-min⁡〖(x)〗 )) (8)

Data set x represents the length of the body lines to a point at which the fuzzy shift applies the same area to the other side. The length of the new points is determined by the function H (x). Using coefficient L at the end of the function, the different sizes of the changes are applied manually. This is the fuzzy function - neurological function obtained from Bell's model to apply to the vessel model:

H(x)=[G(x)×F(X)×L][x/max⁡max⁡〖(x)-min⁡〖(x)〗 〗 +min⁡〖(x)〗 ]

H(x)=[[1/(1+|x/3|^8 )]×[sin⁡((x-min⁡〖(x)〗)/max⁡max⁡〖(x)-min⁡〖(x)〗 〗 ) ]×L][x/max⁡max⁡〖(x)-min⁡〖(x)〗 〗 +min⁡〖(x)〗 ] (9)

Using the obtained membership function, a fuzzy inference system is formed using a membership function H (x) and a simple inference function. Fuzzy relations are applied to any of the vessel body lines based on the customization done. As a result of using the above model, the vessel volume is not altered and the coefficient C_B and C_M are affected positively by adjusting geometrical values.

Q6: Clarify the meaning and significance of the Seakeeping Performance Index (SPI) as the objective function.

R6: The Seakeeping Performance Index (SPI) is a common measure of how well a ship handles rough seas. It calculates the percentage of time that the ship stays within specific motion limits. The SPI depends on assumptions about the frequency of different sea states and the likelihood of different ship speeds and headings. To evaluate the SPI, we predict how the ship will move for each combination of heading, speed, and sea condition. We then compare these predictions to a set of criteria that determine the optimal performance limits for the ship's mission in that particular sea condition.

Q7: The literature review is comprehensive, but consider grouping the studies by themes or methodologies to enhance readability.

R7: thanks for your comment. This is already performed in the manuscript: 

Table 1 Relevant studies on ship hull form optimization. 

Method Conducted by authors

Thin-ship strip theory A. Day & Doctors, 1996; A. H. Day & Doctors, 2001; Hsiung, 1981).

Slender-ship approximation (Yang et al., 2001

Fourier-Kochin flow method Noblesse, 2001; Noblesse & D, 2000

Strip theory B. S. Kim et al., 2020, 2021; Niklas & Karczewski, 2020; Rosa et al., 2021

Potential-flow panel methods based on Rankine sources Campana et al., 1999; Choi et al., 2015; Dejhalla et al., 2001; C.-H. Huang et al., 1998; Lu et al., 2016; Lv et al., 2013; Mittendorf & Papanikolaou, 2021; Peri et al., 2001; Saha et al., 2004; Vu & Nguyen, 2020.

Boundary element method Esmailian et al., 2017; Ghassemi & Zakerdoost, 2017; Zakerdoost et al., 2013; Zakerdoost & Ghassemi, 2018, 2019, 2021, 2023

CFD Campana et al., 2006; Coppedè et al., 2019; Han et al., 2012; He et al., 2011; Hino, 1998, 1999; F. Huang et al., 2015; F. Huang & Yang, 2016; Jeong & Jeong, 2020; H. Kim et al., 2008, 2010, 2012; H. Kim & Yang, 2010; Nazemian & Ghadimi, 2021, 2022b, 2022a; Serani et al., 2022; Tahara, 1998, 2004; Tripathi & Rajagopalan, 2023; Wang & Kim, 2023; Zhang et al., 2018

Q8: Consider presenting the results in a more structured manner, perhaps using subheadings for different aspects of the study.

R8: 

Q9: Discuss the practical implications of the optimized and coefficients for ship design and seakeeping performance.

R9: Naval architects and marine engineers use advanced computational fluid dynamics (CFD) simulations and optimization techniques to calculate and optimize these coefficients for a specific ship design. They can minimize resistance, improve maneuverability, reduce slamming and motions in rough seas, and enhance overall seakeeping performance.

Q10: Summarize the key findings concisely in the conclusion section and emphasize their significance for the field of ship design and optimization.

 R10: The pitch motion can be uncomfortable for passengers and crew and can also affect the stability and performance of the vessel. Hence, decreasing this motion is of great importance in ship design. 

Decreasing MSI will improve the conformability of the crew onboard the vessel.

---

## [Editor Report · Decision Letter 2]

28 Mar 2024

Optimization of ship hull forms by changing CM and CB coefficients to obtain optimal seakeeping performance

PONE-D-23-35446R2

Dear Dr. Ghassemi,

We’re pleased to inform you that your manuscript has been judged scientifically suitable for publication and will be formally accepted for publication once it meets all outstanding technical requirements.

Kind regards,

Dr. S. M. Anas, Ph.D.(Structural Engg.), M.Tech(Earthquake Engg.)

Academic Editor

PLOS ONE

Additional Editor Comments (optional):

Dear Authors,

I hope this email finds you well.

I am writing to inform you that I have reviewed the revised manuscript entitled "Optimization of ship hull forms by changing CM and CB coefficients to obtain optimal seakeeping performance" (Manuscript ID: [PONE-D-23-35446R2]) submitted to PLOS ONE. I am pleased to inform you that the minor revisions suggested by the previous reviewer have been successfully incorporated into the manuscript.

Upon careful consideration of the revisions made by the authors in response to the reviewer's comments, I am satisfied with the thoroughness and quality of the responses. Consequently, I am inclined to recommend acceptance of the manuscript, subject to the approval of the editorial board.

The preliminary assessment of the revised manuscript indicates that it meets the standards of PLOS ONE and contributes significantly to the field. However, the final decision rests with the editorial board, and the manuscript will undergo their evaluation process.

Thank you for your diligence and prompt attention to the reviewer's comments. I appreciate your commitment to improving the quality of your work.

Should you have any questions or require further clarification, please do not hesitate to contact me.

Best regards,

Dr. S. M. Anas

Academic Editor

PLOS ONE